# Hierarchically Organized Latent Modules for Exploratory Search in Morphogenetic Systems

**Mayalen Etcheverry,**<sup>∗</sup> **Clément Moulin-Frier, Pierre-Yves Oudeyer**
Flowers Team
Inria, Univ. Bordeaux, Ensta ParisTech (France)
{mayalen.etcheverry,clement.moulin-frier,pierre-yves.oudeyer}@inria.fr

## Abstract

Self-organization of complex morphological patterns from local interactions is a fascinating phenomenon in many natural and artificial systems. In the artificial world, typical examples of such morphogenetic systems are cellular automata. Yet, their mechanisms are often very hard to grasp and so far scientific discoveries of novel patterns have primarily been relying on manual tuning and ad hoc exploratory search. The problem of automated *diversity-driven* discovery in these systems was recently introduced [26, 62], highlighting that two key ingredients are autonomous exploration and unsupervised representation learning to describe "relevant" degrees of variations in the patterns. In this paper, we motivate the need for what we call *Meta-diversity* search, arguing that there is not a unique ground truth *interesting* diversity as it strongly depends on the final observer and its motives. Using a continuous game-of-life system for experiments, we provide empirical evidences that relying on monolithic architectures for the behavioral embedding design tends to bias the final discoveries (both for hand-defined and unsupervisedly-learned features) which are unlikely to be aligned with the interest of a final end-user. To address these issues, we introduce a novel *dynamic* and *modular* architecture that enables unsupervised learning of a hierarchy of diverse representations. Combined with intrinsically motivated goal exploration algorithms, we show that this system forms a discovery assistant that can efficiently adapt its diversity search towards preferences of a user using only a very small amount of user feedback.

## 1  Introduction

*Self-organisation* refers to a broad range of pattern-formation processes in which globally-coherent structures spontaneously emerge out of local interactions. In many natural and artificial systems, understanding this phenomenon poses fundamental challenges [1]. In biology, *morphogenesis*, where cellular populations self-organize into a structured morphology, is a prime example of complex developmental process where much remains to be assessed. Since Turing's influential paper *"The Chemical Basis of Morphogenesis"* [74], many mathematical and computational models of morphogenetic systems have been proposed and extensively studied [24]. For example, cellular automata abstract models like Conway's Game of Life (GoL), despite their apparent simplicity, generate a wide range of life-like structures ranging from Turing-like patterns reminiscent of animal skins stripes and spots [1] to localized autonomous patterns showing key behaviors like self-replication [38].

A modern goal of morphogenesis research has become the manipulation, exploration and control of self-organizing systems. With the recent advances in synthetic biology and high-precision roboticized experimental conditions, potential applications range from automated material design [72], drug discovery [66] and regenerative medicine [18] to unraveling the chemical origins of life [26, 48]. Yet,

---

<sup>∗</sup>Source code, videos and additional results can be found at `http://mayalenE.github.io/holmes/`

even for simple artificial systems like Conway's Game of Life, we do not fully grasp the mapping from local rules to global structures nor have a clear way to represent and classify the discovered structures. While scientific discoveries of novel patterns have primarily been relying on manual tuning and ad hoc exploratory search, the growing potential of data-driven search coupled with machine learning (ML) algorithms allows us to rethink our approach. In fact, contemporary work opened the path toward three promising directions in applying ML to morphogenesis research, namely in 1) the design of richer computational models, coupling neural networks data-structures [23] with deep-learning techniques to *learn* the update rules of the system, allowing to directly achieve key properties such as self-regeneration and self-replication [49]; 2) by formulating morphological search as a reinforcement learning problem where an artificial agent learns to control the morphologies self-assembly to achieve a certain task such as locomotion [54]; and 3) formulating the problem of automated *diversity-driven* discovery in morphogenetic systems, and proposing to transpose intrinsically-motivated exploration algorithms, coming from the field of developmental robotics, to this new kind of problem [26, 62].

Pure objective-driven search is unlikely to scale to complex morphogenetic systems. It can even be inapplicable when scientists do not know how to characterize desired behaviours from raw observations (reward definition problem), or merely aim to find novel patterns. As an illustration of these challenges, it took 40 years before the first replicator "spaceship" pattern was spotted in Conway's Game of Life. We believe that *diversity-driven* approaches can be powerful discovery tools [26, 62], and can potentially be coupled with objective-driven searches [12, 53, 58]. Recent families of machine learning have shown to be effective at creating *behavioral diversity*, namely Novelty Search (NS) [41, 42] and Quality-Diversity (QD) [16, 58] coming from the field of evolutionary robotics; and intrinsically-motivated goal-directed exploration processes (IMGEP) [2, 21] coming from developmental robotics. A known critical part of these algorithms, is that they require the definition of a *behavioral characterization* (BC) feature space which formalizes the "interesting" degrees of behavioral variation in the environment [57]. So far, this behavior space has either been hand-defined in robotic scenarios [16, 21] or unsupervisedly learned with deep auto-encoders directly from raw observations [15, 39, 51, 55, 56]. While deep auto-encoders have shown to recover the "ground truth" factor of variations in simple generative datasets [5], it is impossible (and not desirable) to recover *all* the degrees of variations in the targeted self-organizing systems.

In this paper, we follow the proposed experimental testbed of Reinke et al. (2020) [62] on a continuous game-of-life system (Lenia, [6]). We provide empirical evidence that the discoveries of an IMGEP operating in a *monolithic* BC space are highly-diverse in that space, yet tend to be poorly-diverse in other potentially-interesting BC spaces. This draws several limitations when it comes to applying such system as a tool for assisting discovery in morphogenetic system, as the suggested discoveries are unlikely to align with the interests of a end-user. How to build an artificial "discovery assistant" learning to generate diverse patterns in the eyes of a future, yet unknown, observer? A central challenge in that direction is to extend the standard notion of *diversity*, where an agent discovers diverse patterns in a monolithic BC space, to what we call *meta-diversity*, where an agent incrementally learns diverse behavioral characterization spaces and discovers diverse patterns within each of them. A second key challenge is to build exploration strategies that can quickly adapt to the preferences of a human end-user, while requiring minimal feedback. To address these challenges, we propose a novel model architecture for unsupervised representation learning with Hierarchically Organized Latent Modules for Exploratory Search, called *HOLMES*. We compare the behavioral characterizations learned by an IMGEP equipped with HOLMES hierarchy of goal space representations to an IMGEP using a single monolithic goal space representation, and the resulting discoveries. We consider two end-user models respectively interested in two types of diversities (diverse spatially localized and diverse turing-like patterns), and show that a monolithic IMGEP will make discoveries that are strongly uneven in relation to these user preferences, while IMGEP-HOLMES is better suited to escape this bias by learning divergent feature representations. Additionally, we show how HOLMES can be efficiently *guided* to drive the search toward those two types of *interesting* diversities with very little amount of (simulated) user feedback.

Our contributions are threefold. We introduce the novel objective of *meta-diversity* search in the context of automated discovery in morphogenetic systems. We propose a *dynamic* and *modular* model architecture for unsupervised learning of *diverse* representations, which, to our knowledge, is the first work that proposes to progressively grow the capacity of the agent visual world model into an organized hierarchical representation. We show how this architecture can easily be leveraged to *drive* exploration, opening interesting perspectives for the integration of a human in the loop.

# 2  Problem Formulation and Motivation for *Meta-Diversity* Search

We summarize the problem of automated discovery of morphogenetic systems as formulated in [62], on a continuous game of life environment example. We identify limits of this formulation and associated approach. The novel process of *meta-diversity* search is proposed within this framework.

**A morphogenetic system: Lenia**  Morphogenetic systems are characterized by an *initial state* ($A^{t=1}$, seed of the system) as well as a set of *local update rules* that are iteratively applied to evolve the state of the system through time ($A^{t=1} \rightarrow \ldots \rightarrow A^{t=T}$). Typically observed from raw images, the emerging patterns depend on a set of *controllable parameters* $\theta$ that, for each experimental run, condition the system initial state and update rules. We use Lenia cellular automaton [6, 7], a continuous generalization to Conway's Game of Life, as testbed. It can generate a wide range of complex behaviors and life-like structures, as testified by the collection of "species" that were manually identified and classified in [6]. To organize the experimental study of exploration algorithms, several restrictions were proposed [62]: the lattice resolution is fixed to $256 \times 256$ and the evolution is stopped after 200 time steps. Moreover, only the system final state is observed ($A^{t=200}$), focusing the search on morphological appearance traits and leaving out dynamical traits (yet see section 4 for side-effect discoveries of interesting dynamics). The controllable parameters include a 7-dimensional set of parameters controlling Lenia's update rule as well as parameters governing the generation of the initial state $A^{t=1}$. Compositional-pattern producing networks (CPPN) [71] are used to produce the initial patterns.

**Automated discovery problem**  The standard automated discovery problem ([62]) consists of generating a maximally-diverse set of observations through sequential experimentation of controllable parameters $\theta$. Each controllable parameter vector $\theta$ generates a rollout $A^{t=1} \rightarrow \ldots \rightarrow A^{t=T}$ of the morphogenetic system leading to an observation $o(\theta) = A^{t=T}$. This observation is encoded as a vector $r = R(o)$ in a BC space representing interesting features of the observation (e.g. based on the color or frequency content of the observation image). Given a budget of N experiments, the objective of the automated discovery problem is to sample a set of controllable parameters $\Theta$ where $\{R(o(\theta))|\theta \in \Theta\}$ maximally covers the BC space.

**Problem definition: Meta-Diversity Search**  The standard automated discovery problem defined above assumes that the intuitive notion of diversity can be captured within a single BC space (what we call a monolithic representation). However, as our results will show, maximizing the diversity in one BC space may lead to poor diversity in other, possibly-interesting, BC spaces. Thus, using a single representation space to drive exploration algorithms limits the scope of their discoveries, as well as the scope of their external evaluation. To address this limit, we formulate the novel process of *meta-diversity* search: in an outer loop, one aims to learn a *diverse* set of behavioral characterizations (called the *meta-diversity*); then in an inner loop, one aims to discover a set of maximally diverse patterns in each of the BC spaces (corresponding to the standard notion of *diversity* in previous work). The objective of this process is to enable continuous seeking of novel niches of diversities while being able to quickly adapt the search toward a new unknown *interesting* diversity. Here, a successful discovery assistant agent is one which can leverage its diverse BCs to specialize efficiently towards a particular type of diversity, corresponding to the initially unknown preferences of an end-user, and expressed through simple and very sparse feedback.

**Proposed approach: IMGEP Agent with modular BC spaces**  A goal-directed intrinsically-motivated exploration process (IMGEP) was used for the parameter sampling strategy in [62]. After an initialization phase, the IMGEP strategy iterates through 1) sampling a goal in a learned BC space $R$, conditioned on the explicit memory of the system $\mathcal{H}$ and based on the goal-sampling strategy $g \sim G(H)$; 2) sampling a set of parameters $\theta$ for the next rollout to achieve that goal, based on its parameter-sampling policy $\Pi = Pr(\theta; g, H)$; 3) let the system rollout, observe outcome $o$, and store the resulting $(\theta, o, R(o))$ triplet in an explicit memory $\mathcal{H}$. However, the behavioral characterization was based on a monolithic representation. Although being learned, this representation was limited to capture diversity in a single BC space and was therefore unable to perform meta-diversity search as defined above. To solve this problem, we introduce a modular architecture where a hierarchy of behavioral characterization spaces is progressively constructed, allowing flexible representations and intuitive guidance during the discovery process.

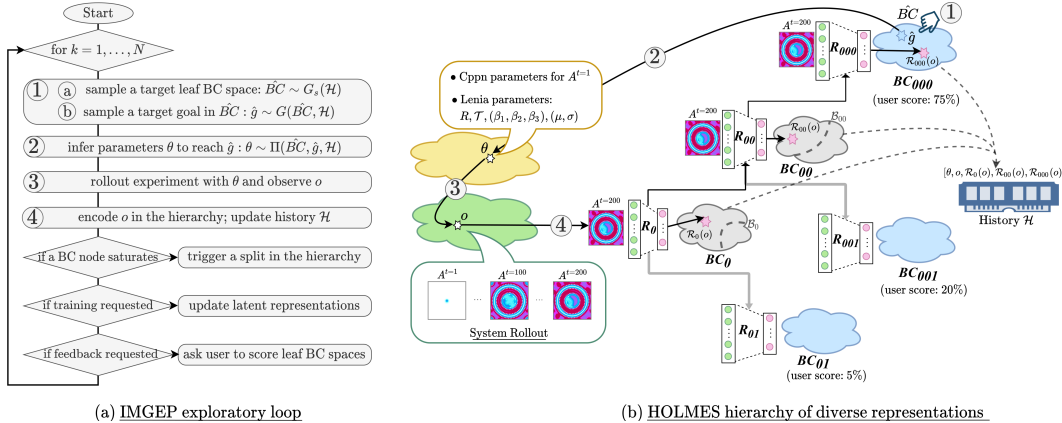

(a) IMGEP exploratory loop　　　　　　　　　　(b) HOLMES hierarchy of diverse representations

Figure 1: IMGEP-HOLMES framework integrates a goal-based intrinsically-motivated exploration process (IMGEP) with the incremental learning of a hierarchy of behavioral characterization spaces (HOLMES). HOLMES unsupervisedly clusters and encodes discovered patterns into the different nodes of the hierarchy of representations. The architecture of HOLMES and clustering part is detailed in section 3.1. The exploratory loop and its interaction with the hierarchy of behavioral characterization (BC) spaces, enabling *meta-diversity search*, is detailed in section 3.2.

## 3 Hierarchically Organized Latent Modules for Exploratory Search

### 3.1 HOLMES: Architecture

HOLMES is a dynamic architecture that "starts small" both on the task data distribution and on the network memory capacity, following the intuition of Elman (1993) [19]. A hierarchy of embedding networks $\mathcal{R} = \{\mathcal{R}_i\}$ is actively expanded to accommodate to the different niches of patterns discovered in the environment, resulting in a progressively deeper hierarchy of specialized BC spaces. The architecture has 4 main components: (i) a base *module* embedding neural network, (ii) a *saturation* signal that triggers the instantiating of new nodes in the hierarchy, (iii) a *boundary* criteria that unsupervisedly clusters the incoming patterns into the different modules; and (iv) a *connection-scheme* that allows feature-wise transfer from a parent module to its children.
We refer to Figure 1 and Figure 6 (section A, appendix) for an illustration of the following.

In this paper, we use a variational auto-encoder (VAE) [33] as base *module*. The hierarchy starts with a single VAE $\mathcal{R}_0$ that is incrementally trained on the incoming data, encoding it into its latent characterization space $BC_0$. A split is triggered when a node of the hierarchy *saturates* i.e. when the reconstruction loss of its VAE reaches a plateau, with additional conditions to prevent premature splitting (minimal node population and a minimal number of training steps). Each time a node gets *saturated*, the split procedure is executed as follows. First, the parameters of the parent $\mathcal{R}_p$ are frozen and two child networks $\mathcal{R}_{p0}$ and $\mathcal{R}_{p1}$ are instantiated with their own neural capacity. Besides, additional learnable layers called *lateral connections* are instantiated between the parent and child VAE feature-maps, drawing inspiration from *Progressive Neural Networks* (PNN) [64]. Here, these layers allow the child VAE to reuse its parent knowledge while learning to characterize novel dissimilar features in its BC space (see section 4.2 and appendix D.3 for an ablation study). Finally, a boundary $\mathcal{B}_p$ is fitted in the parent frozen embedding space $BC_p$ and kept fixed for the rest of the exploration. In this paper, the boundary is unsupervisedly fitted with K-means by assigning two clusters from the points that are currently in the node's BC space. Based on this boundary criteria, incoming observations forwarding through $\mathcal{R}_p$ will be either redirected through the left child $\mathcal{R}_{p0}$ or thought the right child $\mathcal{R}_{p1}$. After the split, training continues: leaf node VAEs as well as their incoming lateral connections are incrementally trained on their own niches of patterns while non-leaf nodes are frozen and serve to redirect the incoming patterns.

In HOLMES, the clustering of the different patterns is central to learn *diverse* BC spaces. The clustering is influenced by the choice of the module and connections training strategy, that determines the latent distribution of patterns in the latent space, and by the clustering algorithm itself. Note that the genericity of HOLMES architecture (agnostic to the base module) allows many other design choices to be considered in future work, and that current choices are discussed in Appendix A.1.

## 3.2 IMGEP-HOLMES: Interaction with the Exploration Process

The goal space of an IMGEP is usually defined as the BC space of interest, with a representation based on a monolithic architecture $\mathcal{R}$ [39, 55]. In this paper, we propose a variant where the IMGEP operates in a hierarchy of goal spaces $\{BC_i\}$, where observations and hence goals are encoded at different levels or granularity, as defined by HOLMES embedding hierarchy $\{\mathcal{R}_i\}$. The exploration process iterates N times through steps 1-to-4, as illustrated in Figure 1. In this section we detail the implementation choices for each step. We refer to algorithm 1 in appendix for a general pseudo-code.

**1)** The goal-sampling strategy is divided in two sub-steps. **a)** Sample a target BC space $\hat{BC_i}$ with a goal space sampling distribution $G_s$. Here, the agent is given an additional degree of control allowing to prioritize exploration in certain nodes of the hierarchy (and therefore on a subset population of patterns). In this paper we considered two setups for the goal space sampling strategy: (i) a *non-guided* variant where the target BC space is sampled uniformly over all the leaf nodes and (ii) a *guided* variant where, after each split in the hierarchy we "pause" exploration and ask for evaluator feedback to assign an *interest score* to the different leaf modules. The guided variant simulates a human observer that could, by visually browsing at few representative images per module and simply "click" and score the leaf nodes with the preferred discoveries. Then during exploration, the agent selects over the leaf goal spaces with softmax sampling on the assigned score probabilities. **b)** Sample a target goal $\hat{g}$ in the selected BC space with a goal sampling distribution $G$. In this paper, we use a uniform sampling strategy: the goal is uniformly sampled in the hypercube envelope of currently-reached goals. Because the volume of the hypercube is larger than the cloud of currently-reached goals, uniform sampling in the hypercube incentivizes to search in unexplored area outside this cloud (this is equivalent to novelty search in the selected BC space). Note that other goal-sampling mechanisms can be considered within the IMGEP framework [2].

**2)** The parameter-sampling strategy $\Pi$ generates the CPPN-generated initial state and Lenia's update rule in 2 steps: (i) given the goal $\hat{g} \in \hat{BC_i}$, select parameters $\hat{\theta}$ in $\mathcal{H}$ whose corresponding outcome $R_i(o)$ is closest to $\hat{g}$ in $\hat{BC_i}$; (ii) mutate the selected parameters by a random process $\theta = \text{MUTATION}(\hat{\theta})$ (see appendix C.2 for implementation details).

**3)** Rollout experiment with parameters $\theta$ and observe outcome $o$.

**4)** Forward $o$ top-down through the hierarchy and retrieve respective embeddings $\{R_i(o)\}$ along the path. Append respective triplets $\{(\theta, o, R_i(o))\}$ to the episodic memory $\mathcal{H}$.

**HOLMES online training.** The data distribution collected by the IMGEP agent directly influences HOLMES splitting procedure and training procedure by determining which nodes get populated and when. In this paper, we incrementally train the goal space hierarchy every $N_T = 100$ exploration step for $E = 100$ epochs on the observations collected in the history $\mathcal{H}$. Importance sampling is used at each training stage, giving more weight to recently discovered patterns. We refer to appendix C.3 for implementation details on HOLMES training procedure.

## 4 Experimental Results

In this section we compare the results of an IMGEP equipped with a goal space based on different types of BCs. We denote **IMGEP-X** an IMGEP operating in a goal space $X$ ($X$ can be e.g. an analytical BC space based on Fourier descriptors, or a modular representation learned by the HOLMES architecture as in section 3.2). In order to evaluate meta-diversity, we make the distinction between the BC used as the goal space of an IMGEP and the BC used for evaluating the level of diversity reached by that IMGEP. For example, we might want to evaluate the diversity reached by **IMGEP-HOLMES** in a BC space based on Fourier descriptors. For quantitative evaluation of the diversity in a given BC, the BC space is discretized with $n = 20$ bins per dimension, and the diversity is measured as the number of bins in which at least one explored entity falls (details are provided in Appendix B.1.2). Each experiment below consists of $N = 5000$ runs starting with $N_{init} = 1000$ initial random explorations runs. For all algorithms, we conduct 10 repetitions of the exploration experiment with different random seeds. Please refer to appendix (sections B and C) for all details on the evaluation procedure and experimental settings. Additionally, the source code and full database of discoveries are provided on the project website.

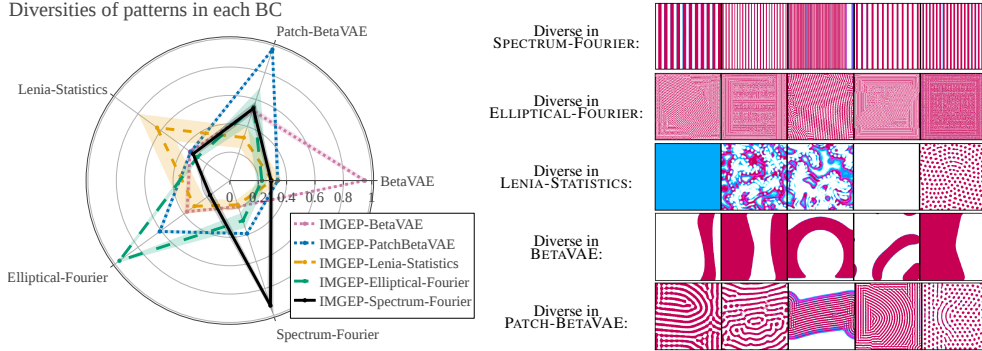

Figure 2: Although IMGEPs succeed to reach a high-diversity in their respective BC space, they are poorly-diverse in all the others. (left) Diversity for all IMGEP variants measured in each analytic BC space. For better visualisation the resulting diversities are divided by the maximum along each axis. Mean and std-deviation shaded area curves are depicted. (right). Examples of patterns discovered by the IMGEPs that are consider diverse in their respective BC space. See Appendix B.1.1 for details.

## 4.1 Does maximizing diversity in a given BC space lead to high diversity in other BC spaces?

We construct 5 BC spaces with 8 dimensions each, among which 3 representations are predefined and 2 are pre-learned on a database of Lenia patterns. The exploration discoveries of an IMGEP equipped with the different BCs as (fixed) goal space are evaluated and compared in figure 2. The two first BCs rely on Fourier descriptors, intended to characterize the frequencies of textures (Spectrum-Fourier) and shape of closed contours (Elliptical-Fourier [36]), typically used in cellular-automata [47] and biology [14, 80]. The third BC relies on statistical features that were proposed in the original Lenia paper [6] to describe the activation of patterns. The fourth uses a set of features unsupervisedly learned by a $\beta$-VAE [5] on a large database of Lenia patterns, as proposed in [62]. Because $\beta$-VAE can poorly represent high-frequency patterns, another variant trained on cropped patches is proposed.

**Limits of monolithic goal spaces**    The results of Figure 2 suggest that, if we could have a theoretical BC model that aligns with what a user considers as diverse under the form of a goal space, the IMGEP should be efficient in finding a diversity of patterns in that space. In practice however, constructing such a BC is very challenging, if not infeasible. Each BC was carefully designed or unsupervisedly learned to represent what could be "relevant" factors of variations and yet, the IMGEP seems to exploit degrees of variations that might not be aligned with what we had in mind when constructing such BCs. Spectrum-Fourier is a clear example that was constructed to describe textures (in a general sense) but where the discoveries exhibit only vertical-stripe patterns with all kind of frequencies.

## 4.2 What is the impact of modularity in incremental learning of goal space(s)?

**Baselines**    We compare **IMGEP-VAE** which uses a monolithic VAE as goal space representation to **IMGEP-HOLMES** which is defined in section 3. HOLMES expansion is stopped after 11 splits (resulting in a hierarchy of 23 VAEs) and uses small-capacity modules and connections, such that its final capacity is still smaller than the monolithic VAE. Other variants for the monolithic architecture and training strategies are considered in Appendix D.2.

**Learning to characterize different niches**    We use representational similarity analysis (RSA) [35] to quantify how much the representations embeddings (encoded behaviors) evolve through the exploration inner loop (Figure 3). Different metrics have been proposed to compute the representational similarity, here we use the linear centered kernel alignment (CKA) index [34] (see Appendix B.2). Results show that the learned features of the monolithic VAE stop evolving after only 2 training stages, i.e. 200 explored patterns. This suggests that, even though the VAE is *incrementally trained* during exploration (at the difference of the pretrained variants in section 4.1), it still fails to adapt to the different niches of patterns which will lead to limited discoveries. However, RSA results suggest that HOLMES succeeds to learn features that are highly dissimilar from one module to another, which allows to target discovery of a *meta-diversity*. An ablation study (see section  D.3 of appendix) highlighted the importance of HOLMES *lateral connections* to escape the VAE learning biases.

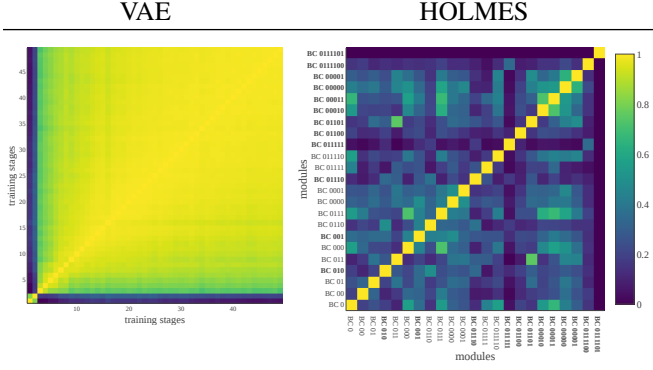

VAE&emsp;&emsp;&emsp;&emsp;&emsp;HOLMES

Figure 3: RSA similarity index between 0 (dark blue, not similar at all) and 1 (yellow, identical). (VAE) representations are compared in time between the different training stages. (HOLMES) representations are compared at the *end* of exploration between the different modules. Leafs are depicted in bold and modules are ordered by their creation time (left-to-right in x-axis). See Appendix D.1 for full temporal analysis and statistical results.

**Learning to explore different niches** &emsp; Qualitative browsing through the discoveries confirms that IMGEP-HOLMES is able to explore diverse niches, allowing the discovery of very interesting behaviors in Lenia from quite unexpected types of patterns. To further investigate these discoveries, the exploration was prolonged for 10000 additional runs (without expanding more the hierarchy). Not only many of the "lifeform" patterns of the *species* identified in [6] were discovered, but it allowed to assist to the *birth* of these creatures from fluid-like structures that have shown capable of *pattern-emission* (behavior which was, to our knowledge, never discovered in 2D Lenia). Example of our discoveries can be seen in Figure 4 and on the website `http://mayalenE.github.io/holmes/`.

## 4.3 &emsp; Can we drive the search toward an *interesting* diversity?

Two categories of patterns have been extensively studied in cellular-automata, known as Spatially-Localized Patterns (SLP) and Turing-Like Patterns (TLP) (Figure 4). We investigate if our *discovery assistant* search can be guided to specialize toward a diversity of either SLPs or TLPs. For experimentation, we propose to use a *simulated* user, preferring either SLPs or TLPs, and a *proxy* evaluation of diversity tailored to these preferences. For *guidance*, the classifiers of "animals" (SLP) and "non-animals" (TLP) from [62] are used to score the different nodes in IMGEP-HOLMES with the number of SLP (or TLP) that fall in that node at split time (see section 3.2). This simulates preferences of a user toward either SLPs or TLPs, who would use a GUI to score the patterns found in each leaves of IMGEP-HOLMES. The total number of user interventions is 11 (one per split) with, for each intervention, an average of 6 "clicks" (scores), which represents very sparse feedback. For evaluation, an experiment with a human evaluator has been conducted for selecting the BC (among the 5 proposed in section 4.1) that correlates the most with the evaluator judgement of what represents a diversity of SLP and a diversity of TLP. $BC_{\text{ELLIPTICAL-FOURIER}}$ was designed as the best proxy space for evaluating the diversity of SLP (98% agreement score) and $BC_{\text{LENIA-STATISTICS}}$ was designated for TLP (92% agreement). Experiment details are provided in appendix B.3.

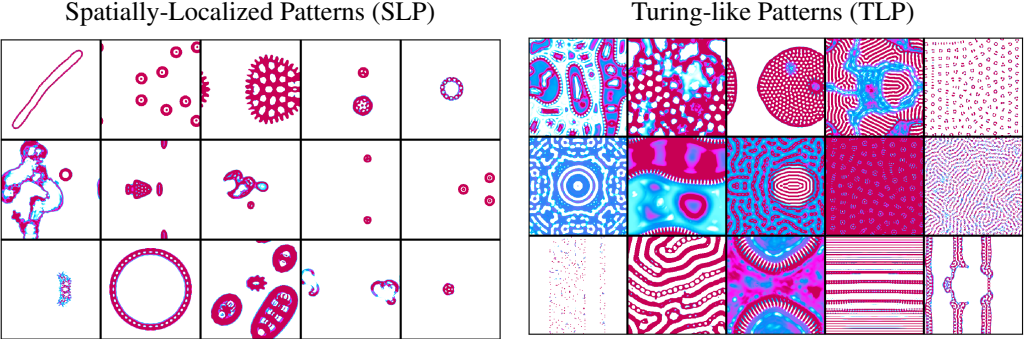

Spatially-Localized Patterns (SLP)&emsp;&emsp;&emsp;&emsp;Turing-like Patterns (TLP)

Figure 4: (Left) SLP are autonomous stable patterns, that show interesting behaviors such as locomotion and metamorphosis(shape-shifting). (Right) TLP patterns are characterized by an unlimited spatial growth resembling reaction-diffusion pattern-formation of fronts, spirals, stripes and dissipative solitons. The displayed patterns where *autonomously* discovered in Lenia [6] by IMGEP-HOLMES (without guidance) and, considered by us (human evaluator), as *interesting*.

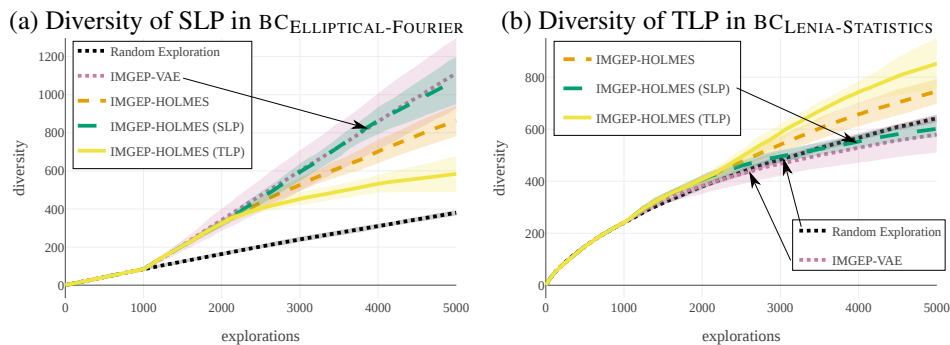

Figure 5: Depicted is the diversity discovered by the algorithms during exploration. The discovered patterns classified as SLP in (a) (resp TLP in (b)) are projected in $BC_{\text{ELLIPTICAL-FOURIER}}$ space in (a) (resp $BC_{\text{LENIA-STATISTICS}}$ in (b)), where the binning-based measure is used. Mean and std-deviation shaded area curves are depicted.

**Baselines**  Non-guided IMGEP variants (section 4.2) are compared with guided IMGEP variants. The results are compared to Random Exploration baseline (where parameters $\theta$ are randomly sampled for the 5000 explorations) which serves as reference for the *default* diversity found in Lenia (and by all algorithms during the first 1000 explorations).

**Results**  The results in Figure 5 show that the bias of the monolithic VAE allows IMGEP-VAE to find a high diversity of SLPs but leads to poor diversity of TLPs. When non-guided, IMGEP-HOLMES finds a higher diversity than Random Exploration both for SLPs and TLPs. When guided, IMGEP-HOLMES can even further increase the diversity in the category of interest.

## 5    Related Work

**Machine Learning for Automated Discovery in Science**  Many work successfully applied ML for discovery in chemistry [17, 25, 60], physics [4, 46] and biology [27, 32]. However, they focus the search on predefined structures with target properties and not, to our knowledge, on diversity search.

**Diversity Search**  Diversity-driven approaches in ML are presented in the introduction. In the QD literature, the work of Pugh et al. (2016) [59] relates to what we frame as *meta-diversity* search. In a maze environment where 2 sets of hand-defined BC are provided (one for agent position and one for agent direction), they show that driving the search with the two BC sets simultaneously leads to higher probability to discover *quality* behavior (meaning here solving the task) than with a single *unaligned* BC. However, BCs are predefined and fixed, limiting the generalisation to complex environments.

**State Representation Learning**  Many approaches have been proposed for state representation learning in reinforcement learning [43]. As presented in the introduction, goal-directed approaches generally rely on deep generative models such as VAEs [51, 55]. Others tune the representation to achieve target properties at the feature level such as disentanglement [29] or linear predictability [76]. This includes coupling with predictive forward and inverse models [28, 29, 31, 53, 75, 79] or priors of independent controllability [39, 73]. However, they all rely on a single embedding space, where all the observed instances are mapped to the same set of features.

**Continual Unsupervised Representation Learning**  Recent work also proposed to dynamically expand the network capacity of a VAE [40, 45, 61]. Similarities and differences with HOLMES are discussed in Appendix E. However, these approaches were applied to passively observed datasets, either targeting unsupervised clustering of sequentially-received class of images or disentanglement of factors of variations in generative datasets.

**Interactive Exploration of Patterns**  Interactive evolutionary computation (IEC) [37, 68, 69] aims to integrate external human feedback to explore high complexity pattern spaces, via intuitive interfaces. However, the user must provide feedback at each generation by individually selecting interesting patterns for the next generation; whereas our framework requires much sparser feedback.

# 6 Discussion

As stated in the Introduction, our contributions in this paper are threefold. First, in section 2, we introduced the novel objective of *meta-diversity* search in the context of automated discovery in morphogenetic systems. Then, in section 3, we proposed a *dynamic* and *modular* model architecture for meta-diversity search through unsupervised learning of *diverse* representations. Finally, in section 4, we showed that search can easily be guided toward complex pattern preferences of a simulated end user, using very little user feedback.

To our knowledge, HOLMES is the first algorithm that proposes to progressively grow the capacity of the agent visual world model into an organized hierarchical representation. There are however several limitations to be addressed in future works. The architecture remains quite *rigid* in the way it is isolating the different niches of patterns (binary tree with frozen boundaries) whereas other approaches, further leveraging human feedback, could be envisaged.

The question whether *machines* can really help scientists for crucial discoveries in Science, although appealing, is still an open question [3]. We believe that machine learning algorithms integrating flexible modular representation learning with intrinsically-motivated goal exploration processes for *meta-diversity* search are very promising directions. As an example, despite the limitations mentioned above, IMGEP-HOLMES was able to discover many types of solitons including unseen pattern-emitting lifeforms in less than 15000 training steps without guidance, when their existence remained an open question raised in the original Lenia paper [6].

## Broader Impact Statement

We introduced methods that can be used as tools to help human scientists discover novel structures in complex dynamical systems. While experiments presented in this article were performed using an artificial system (continuous cellular automaton), they also target to be used for automated discovery of novel structures in fields ranging from biology to physics. As an example, Grizou et al. [26] recently showed how IMGEPs can be used to automate chemistry experiments addressing fundamental questions related to the origins of life (how oil droplets may self-organize into proto-cellular structures), leading to new insights about oil droplet chemistry. As experiments in Grizou et al. used a single pre-defined BC, one can expect that the new approach presented in this paper may boost the efficiency of its use in bio-physical systems, that could include systems related to design of new materials or new drugs. As a tool enabling scientist to better understand the space of dynamics of such systems, we believe it could help them better understand how to leverage such dynamics for societally useful purposes, and avoid negative effects, e.g. due to unpredicted self-organized dynamics.

However, technological and scientific discoveries might have a considerable impact in modern societies. Introducing machine decisions in the process should therefore be done with great responsibility, taking care of carefully identifying and balancing the biases inherent to any ML algorithms. The methods proposed in this paper constitute a first step in this direction by quantitatively measuring the influence of biases, in both predefined and learned BC spaces, on the algorithm discoveries. With an increasing interest in ML for automated discovery, it will be fundamental to to improve and extend these methods in the near future.

Besides, by releasing our code, we believe that we help efforts in reproducible science and allow the wider community to build upon and extend our work in the future.

## Acknowledgements and Disclosure of Funding

We would like to thank Chris Reinke and Bert Chan for useful inputs and discussions for the paper, as well as Jonathan Grizou for valuable suggestions.

The authors acknowledge support from the Human Frontiers Science Program (Collaborative Research Grant RGP0018/2016) and from Inria.

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
