[Supplementary Material]

# Supplementary Material

This supplementary material provides implementation details, hyper-parameters settings, additional results and visualisations.

- Section A presents a focus on the design choices we use for IMGEP-HOLMES
- Section B provides implementation details for the main paper evaluation procedure
  - B.1: Quantitative evaluation of diversity
  - B.2: Quantitative evaluation of Representational Similarity
  - B.3: Human-evaluator selection of the BC spaces for evaluating SLP and TLP diversity
- Section C provides all necessary implementation details for reproducing the main paper experiments
  - C.1: Lenia environment settings
  - C.2: Parameter-sampling policy $\Pi$ settings for Lenia's initial state and update rule
  - C.3: Settings for training the BC spaces in IMGEP-VAE and IMGEP-HOLMES
- Section D provides additional results that complete the ones from the main paper
  - D.1: Complete RSA analysis of the hierarchy of behavioral characterizations learned in HOLMES
  - D.2: Additional IMGEP baselines with a monolithic BC space are compared
  - D.3: Ablation study of the impact of the lateral connections in HOLMES
- Section E discusses the comparison of HOLMES with other model-expansion architectures
- Section F provides qualitative visualisations of the hierarchical trees that were autonomously constructed by the different IMGEP-HOLMES variants.

**Source code:**  Please refer to the project website `http://mayalenE.github.io/holmes/` for the source code and complete database of discoveries for our experiments.

# A Focus on IMGEP-HOLMES

**Algorithm 1: IMGEP-HOLMES pseudo-code.** Please refer to section 3 of the main paper for the step-by-step implementation choices and to section C in apppendix for the implementation details.

---

**Algorithm 1** IMGEP-HOLMES

---

**Inputs:** Parameter-sampling policy $\Pi$
Initialize root representation $\mathcal{R} = \{\mathcal{R}_0\}$
**for** $k \leftarrow 1$ **to** $N$ **do**
    **if** $k < N_{init}$ **then** # *initial random iterations*
        Sample $\theta \sim \mathcal{U}(\Theta)$
    **else** # *goal-directed iterations*
        Sample a target BC space $\hat{BC} \sim G_s(\mathcal{H})$
        Sample a goal $\hat{g} \sim G(\hat{BC}, \mathcal{H})$ in $\hat{BC}$
        Choose $\theta \sim \Pi(\hat{BC}, \hat{g}, \mathcal{H})$
    Rollout experiment with $\theta$ and observe $o$

    # *Encode reached goals in the hierarchy*
    Start with root node $i \leftarrow 0, parent(i) = \emptyset$
    **while** $i$ exists in the hierarchy (until leaf) **do**
        $r_i = \mathcal{R}_i(o, \mathcal{R}_{parent(i)}(o))$
        Append $(\theta, o, r_i)$ to the history $\mathcal{H}$
        $i \leftarrow ic, c = \mathcal{B}_i(r_i)$ # *go to left or right child*

    # *Augment representational capacity*
    **if** a BC space $BC_i$ is *saturated* **then**
        Freeze $\mathcal{R}_i$ weights
        Define a boundary $\mathcal{B}_i : BC_i \rightarrow \{0, 1\}$
        Instantiate child modules $\mathcal{R}_{i0}$ and $\mathcal{R}_{i1}$
        # *Project past discoveries to children BCs*
        **for** $(\theta, o, r) \in \mathcal{H}[BC_i]$ **do**
            $\mathcal{R}_j \leftarrow \mathcal{R}_{ic}, c = B_i(r)$
            Append $(\theta, o, \mathcal{R}_j(o))$ to $\mathcal{H}[j]$

# *Periodically train HOLMES*
**if** training requested **then**
    **for** E epochs **do**
        Train the hierarchy $\mathcal{R}$ on observations in $\mathcal{H}$
        with importance sampling
    # *Update the database of reached goals*
    **for** $i \in$ hierarchy **do**
        **for** $(\theta, o, r) \in \mathcal{H}[i]$ **do**
            $\mathcal{H}[i][r] \leftarrow \mathcal{R}_i(o)$

# *Ask for user feedback*
**if** user feedback requested **then**
    Ask user to score leaf BC spaces
    Update $G_s$ with assigned scores

---

## A.1 Design choices in HOLMES

Figure 6: Focus on the different design choices made for the HOLMES architecture. (Left) Each module uses a VAE [33] as the base architecture, where the embedding $R_i$ is coupled to a decoder $D_i$ ($D_i$ is not shown on the left panel for readability). All non-leaf node VAEs are frozen as well as their incoming lateral connections (light grey). The leaf nodes are incrementally trained on their own niches of patterns (represented as colored squares above the embeddings) defined by the boundaries fitted at each node split (curved dotted lines in each BC space, represented as clouds). (Right) $R_{011}$ is trained to encode new information in a latent representation $r_{011}$ (plain vertical arrow) by learning to reuse its parent knowledge via the lateral connections (dotted arrows, denoted as l_f, gfi_c, lfi_c, recon_c).

While the global architecture is generic and numerous design choices can be made, this section details the practical implementation for the *modules*, *connection scheme*, and *splitting criteria* used in this paper. We summarize those components in Figure 6.

**Choice for the base module**    Each module has an embedding network $R_i$ that maps an observation $o$ to a low-dimensional vector $r = R(o)$. To learn such embedding, we rely on a variational autoencoder network [33] for the base module. The encoder network $R_i : q_\phi(r|x)$ is coupled with a decoder network $D_i : p_\theta(x|r)$ that enables a generative process from the latent space, and the networks are jointly trained to maximize the marginal log-likelihood of the training data with a regularizer on the latent structure.
The training loss is $\mathcal{L}_{\text{VAE}}(\theta, \phi; \mathbf{x}, \mathbf{r}) = \underbrace{\mathbb{E}_{\hat{p}(\mathbf{x})} \left( \mathbb{E}_{q_\phi(\mathbf{r}|\mathbf{x})} \left( - \log p_\theta(\mathbf{x}|\mathbf{r}) \right) \right)}_{a} + \underbrace{\mathbb{E}_{\hat{p}(\mathbf{x})} \left( D_{\text{KL}} \left( q_\phi(\mathbf{r}|\mathbf{x}) || p(\mathbf{r}) \right) \right)}_{b},$
where (a) represents the expected reconstruction error (computed with binary cross entropy) and (b) is the regularizer KL divergence loss of the approximate diagonal Gaussian posterior $q_\phi(r|x)$ from the standard Gaussian prior $p(z) = \mathcal{N}(0, I)$. Please note that input observations are partitioned between the different nodes in HOLMES, therefore each module VAE is trained only on its niche of patterns. Only the encoder network $R_i$ is kept in IMGEP-HOLMES (Algorihm 1), therefore other choices for the base module and training strategy could be envisaged in future work, for instance with contrastive approaches instead of generative approaches.

**Choice for the connection scheme**    The connection scheme takes inspiration from *Progressive Neural Networks* (PNN) [64]), where transfer is enabled by connecting the different modules via learned *lateral connections*. To mitigate the growing number of parameters, we opted for a sparser connection scheme that in [64]. The connection scheme is summarized in Figure 6. Connections are only instantiated between a child and its parent (hierarchical passing of information). Connections are only instantiated between a reduced number of layers (denoted as l_f, gfi_c, lfi_c, recon_c in the figure). We hypothesize that transfer is beneficial in the decoder network so a child module can reconstruct "as well as" its parent, however connections are removed between encoders as new complementary type of features should be learned. We preserve the connections only at the local feature level, as the CNN first layers tend to learn similar features [78]. Connections between linear layers are defined as linear layers and connections between convolutional layers are defined as convolutions with $1 \times 1$ kernel. At each connection level, the output of the connection is summed to the current feature map in the VAE. Other connection schemes could be envisaged in future work, for instance with FiLM layers [55] (feature-wise affine transformation instead of sum) which have recently been proposed for vision models.

**Choice for the splitting criteria**    There are two main choices: *when* to split a node and *how* to redirect the patterns toward either the left or right children. For both, we opted for simple design choices that allow the split to be unsupervisedly and autonomously handled during the exploration loop. We trigger a split in a node when the reconstruction loss of its VAE reaches a plateau, with additional conditions to prevent premature splitting (minimal node population and minimal number of training steps) or to limit the total number of splits. When splitting a node, we use K-means algorithm in the embedding space to fit 2 clusters on the points that are currently in the node. This generates a boundary in the latent space of the node, that we keep fixed for the rest of the exploration loop. Again, many other choices could be envisaged in future work, for instance by including human feedback to fit the boundary or with more advanced clustering algorithms.

# B    Complete Description of the Evaluation Procedure

## B.1    Evaluation of diversity

### B.1.1    Construction of 5 analytic BC spaces

This section details the 5 BC spaces introduced in section 4.1 of the main paper. Each set of BC features relies either on *engineered* representation based on existing image descriptors from the literature or on *pretrained* representations unsupervisedly learned on Lenia patterns. Those BCs were constructed to characterize different *types* of diversities in the scope of evaluating *meta*-diversity as defined in section 2, but obviously many others could be envisaged. The 5 BC models are provided with the source code of this paper.

Each set of BC features is defined by a mapping function $BC_X : o \in [0,1]^{256 \times 256} \mapsto \hat{z} \in [0,1]^8$ where $X$ is the corresponding BC space, $o$ is a Lenia pattern and $\hat{z}$ represents its 8-dimensional behavioral descriptor in the corresponding BC space.

We denote $\mathcal{D}_{ref}$ an external dataset of 15000 Lenia patterns. The patterns in $\mathcal{D}_{ref}$ were randomly collected from prior exploration experiments in Lenia, experiments that include different random seeds and different exploration variants and comport 50% SLPs and 50% TLPs. $\mathcal{D}_{ref}$ is a large database that is intended to cover a diversity of patterns orders of magnitude larger than what could be found in any single algorithm experiment, and that we use as reference dataset to construct and normalize the different evaluation BC spaces.

**Spectrum-Fourier**    The 2-dimensional discrete Fourier transform is a mathematical method that projects an image (2D spatial signal) into the frequency domain, from which frequency characteristics can be extracted and used as texture descriptors [70]. Applications range from material description [50], leaf texture description in biology [13] and rule classification in cellular automata [47].

The construction of $BC_{\text{SPECTRUM-FOURIER}}$ is summarized in Figure 7 and follows the below procedure:

1. The 2D Fast Fourier Transform transforms the image $o = f(x,y)$ into the $u, v$ frequency domain function $F$, the zero-frequency component is shifted to the center of the array and the power specrum $PS$ (or power spectral density) is computed:

$$F(u,v) = \frac{1}{256 \times 256} \sum_{x=0}^{255} \sum_{y=0}^{255} f(x,y) \exp^{-j2\pi \frac{ux}{256} \frac{vx}{256}}$$

$$F(u,v) \leftarrow Roll(F(u,v), (\frac{256}{2}, \frac{256}{2}))$$

$$PS(u,v) = Real(F(u,v))^2 + Imaginary(F(u,v))^2$$

2. The power spectrum is filtered to keep only the lower half (symmetry property of the FFT) and the significant values:

$$PS(u,v) = \{PS(u,v), 0 \le u \le \frac{256}{2}, -\frac{256}{2} \le v \le \frac{256}{2} - 1\}$$

$$PS(u,v) = 0 \; if \; PS(u,v) < mean(PS(u,v))$$

3. The power spectrum is partitioned into 20 ring-shaped sectors:

$$\left[ R_i = \{PS(u,v)|r_1^2 \le u^2 + v^2 \le r_2^2\} \; with \; (r_1, r_2) = (\frac{i}{20} \times \frac{256}{2}, \frac{i+1}{20} \times \frac{256}{2}); \; for \; i \in [0..19] \right]$$

4. A 40-dimensional feature vector (FV) representing radially-aggregating measures (mean $\mu_i$ and standard deviation $\sigma_i$ of each sector) is extracted:

$$FV(o) = [\mu_1, \sigma_1, \ldots, \mu_{20}, \sigma_{20}],$$
$$\text{where } \mu_i = mean(PS[R_i]), \sigma_i = std(PS[R_i])$$

5. The 40-dimensional feature vector $FV$ is projected into a normalized 8-dimensional behavioral descriptor $\hat{z}$ using a transformation $\hat{T} : FV \mapsto \hat{z}$. $\hat{T}$ is constructed with Principal

Component Analysis (PCA) [77] dimensionality reduction on $\mathcal{D}_{ref}$:

$$X_{ref} = \{FV(o), o \in \mathcal{D}_{ref}\}$$

Fit a PCA with 8 components on $X_{ref}$, $PCA : FV \in \mathbb{R}^{40} \mapsto z \in \mathbb{R}^8$

$$z_{ref} = PCA(X_{ref}), z_{min} = percentile(z_{ref}, 0.01), z_{max} = percentile(z_{ref}, 99.9))$$

$$\hat{T} : FV \mapsto \hat{z} = \frac{PCA(FV) - z_{min}}{z_{max} - z_{min}}$$

6. $\text{BC}_{\text{SPECTRUM-FOURIER}}(o) = \hat{T} \circ FV(o)$

Figure 7: Construction of SPECTRUM-FOURIER analytic space. See text for details. Please note that for visualisation purposes: (left) the original image is colorized but is originally a $256 \times 256$ grayscale image; (step 1-2-3) the power spectrum is depicted in logarithmic scale.

**Elliptical-Fourier** Elliptical Fourier analysis (EFA) [36] is a mathematical method for contour description which has been widely-used for shape description in image processing [63]. These descriptors have been applied to morphometrical analysis in biology [44], for instance to characterize the phenotype of plants leaf and petal contours [52] or anatomical shape changes [10, 22].

A closed contour $\{x_p, y_p\}_{p=1}^{K}$ (K points polygon) can be seen as a continuous periodic function of the *length* parameter $T = \sum_{p=1}^{K} \Delta t_p$ where $t_p$ is the distance from the $p-1^{th}$ to the $p^{th}$ point. Therefore it can be represented as a sum of cosine and sine functions of growing frequencies (harmonics) under Fourier approximation. Each harmonic is an ellipse which is defined by 4 coefficients $a, b, c, d$.

The construction of $\text{BC}_{\text{ELLIPTICAL-FOURIER}}$ is summarized in Figure 8 and follows the below procedure:

1. Binarize the image $o_{binary} = o > 0.2$ and extract the external contour as the a list of the (x,y) positions of the pixels that make up the boundary using OpenCV function `contour = cv2.findContours(o_binary, cv2.RETR_EXTERNAL, cv2.CHAIN_APPROX_SIMPLE)`

2. Extract the set of $\{a_n, b_n, c_n, d_n\}_{n=1}^{N}$ coefficients for a series of N ellipses (N=25) from the x- and y-deltas ($\Delta x_p$ and $\Delta y_p$) between each consecutive point p in the K points polygon:

$$a_n = \frac{T}{2n^2\pi^2} \sum_{p=1}^{K} \frac{\Delta x_p}{\Delta t_p} \left[ \cos\frac{2n\pi t_p}{T} - \cos\frac{2n\pi t_{p-1}}{T} \right]$$

$$b_n = \frac{T}{2n^2\pi^2} \sum_{p=1}^{K} \frac{\Delta x_p}{\Delta t_p} \left[ \sin\frac{2n\pi t_p}{T} - \sin\frac{2n\pi t_{p-1}}{T} \right]$$

$$c_n = \frac{T}{2n^2\pi^2} \sum_{p=1}^{K} \frac{\Delta y_p}{\Delta t_p} \left[ \cos\frac{2n\pi t_p}{T} - \cos\frac{2n\pi t_{p-1}}{T} \right]$$

$$d_n = \frac{T}{2n^2\pi^2} \sum_{p=1}^{K} \frac{\Delta y_p}{\Delta t_p} \left[ \sin\frac{2n\pi t_p}{T} - \sin\frac{2n\pi t_{p-1}}{T} \right]$$

3. The coefficients are standardized (i.e. made invariant to size, rotation and shift):
$$\begin{bmatrix} a_n^* & b_n^* \\ c_n^* & d_n^* \end{bmatrix} = \frac{1}{L} \begin{bmatrix} \cos\phi & \sin\phi \\ -\sin\phi & \cos\phi \end{bmatrix} \begin{bmatrix} a_n & b_n \\ c_n & d_n \end{bmatrix} \begin{bmatrix} \cos N\theta & \sin N\theta \\ -\sin N\theta & \cos N\theta \end{bmatrix}, \quad \text{where} \quad L = \sqrt{[(A_0 - x_m)^2 + (C_0 - x_m)^2]}, (A_0, C_0) \text{ is the center of the } 1^{st} \text{ harmonic ellipse, } (x_m, y_m)$$

is the location of the modified starting point (on the major axis of the ellipse), $\theta = \frac{2\pi t_m}{T}$ and $\phi = \tan^{-1}\frac{y_m - C_0}{x_m - A_0}$ (angle between the major axis of the ellipse and xaxis).

4. The 100-dimensional feature vector $FV = \{a_n^*, b_n^*, c_n^*, d_n^*\}_{n=1}^{25}$ is projected into a normalized 8-dimensional behavioral descriptor using a transformation $\hat{T} : FV \mapsto \hat{z}$. $\hat{T}$ is constructed with Principal Component Analysis (PCA) dimensionality reduction on $\mathcal{D}_{ref}$ (similar procedure as in point 5 of $\mathrm{BC_{SPECTRUM\text{-}FOURIER}}$).

5. $\mathrm{BC_{ELLIPTICAL\text{-}FOURIER}}(o) = \hat{T} \circ FV(o)$

Figure 8: Construction of ELLIPTICAL-FOURIER analytic space. See text for details. (step 1) The contour depicted in green is extracted with OpenCV `findContours()` function (step 2) The contours depicted in red are reconstructed from the EFA coefficients (like in other Fourier series transforms the shape signal can be approximated by summing the harmonics [36]).

**Lenia-Statistics**  The original Lenia paper proposes several measures for statistical analysis of the Lenia patterns (section 2.4.2 in [6]), also defined in Reinke et al. (2020) (section B.3 in [62]).

$\mathrm{BC_{LENIA\text{-}STATISTICS}}$ is constructed on top of these measures according to the below procedure:

1. Among all the statistical measures proposed in [6] we selected the 17 measures that are time-independent, i.e. that can be computed from the final Lenia pattern $o = I(x, y)$, namely:
   - the activation mass $m = \frac{1}{256 \times 256} \sum_{(x,y) \in I} I(x, y)$
   - the activation volume $V_m = \frac{1}{256 \times 256} \sum_{(x,y) \in I} \delta_{I(x,y) > \epsilon}$ ($\epsilon = 10^{-4}$)
   - the activation density $\rho_m = \frac{m}{V_m}$
   - the centeredness of the activation mass distribution
     $C_m = \frac{1}{m} \sum_{(x,y) \in I} w_{xy} \cdot I(x - \bar{x}_m, y - \bar{y}_m)$ where $(\bar{x}_m, \bar{y}_m)$ is the activation centroid
     and $w_{x,y} = \left(1 - \frac{d(x,y)}{\max\limits_{x,y} d(x,y)}\right)^2$ with $d(x, y) = \sqrt{(x - \bar{x}_m)^2 + (y - \bar{y}_m)^2}$
   - the 8 invariant image moments by Hu [30]
   - the 5 extra invariant image moments by Flusser [20]

2. The 17-dimensional feature vector $FV = [m, V_m, \rho_m, C_m, hu_1, \ldots, hu_7, flusser_8, \ldots, flusser_{13}]$ is projected into a normalized 8-dimensional behavioral descriptor using a transformation $\hat{T} : FV \mapsto \hat{z}$. $\hat{T}$ is constructed with Principal Component Analysis (PCA) dimensionality reduction on $\mathcal{D}_{ref}$ (similarly than for $\mathrm{BC_{SPECTRUM\text{-}FOURIER}}$).

3. $\mathrm{BC_{LENIA\text{-}STATISTICS}}(o) = \hat{T} \circ FV(o)$

**BetaVAE**  Reinke et al. (2020) [62] propose to train a $\beta$-VAE [5] on a large database of Lenia pattterns and to reuse the learned features as behavioral descriptors for the analytic BC space.

$\mathrm{BC_{BETAVAE}}$ is constructed according to the below procedure :

1. A $\beta$-VAE with 8-dimensional latent space is instantiated with the architecture detailed in table 1.

2. The construction of the training dataset, training procedure and hyperparameters follows [62]:

- The $\beta$-VAE is trained on an external database $\mathcal{D}_{ref}^{(big)}$ of 42500 Lenia patterns (with 50% SLP and 50% TLP, 37500 as training set, 5000 as validation set) which were randomly collected from independent previous experiments (with the same procedure than $\mathcal{D}_{ref}$).
- The $\beta$-VAE is trained for more than 1250 epochs with hyperparameters $\beta = 5$, Adam optimizer ($lr = 1e-3$, $\beta_1 = 0.9$, $\beta_2 = 0.999$, $\epsilon = 1e-8$, weight decay=1e−5) and a batch size of 64.
- The network weights which resulted in the minimal validation set error during the training are kept.

3. The resulting pretrained encoder serves as mapping function from a Lenia pattern $o$ to a 8-dimensional feature vector $FV(o) = [z_1, z_2, z_3, z_4, z_5, z_6, z_7, z_8]$

4. Similarly to the other analytic BC spaces in this paper, we use the reference dataset $\mathcal{D}_{ref}$ to normalize the 8-dimensional behavioral descriptors between $[0, 1]$:

$$z_{ref} = \{FV(o), o \in \mathcal{D}_{ref}\}, z_{min} = percentile(z_{ref}, 0.01), z_{max} = percentile(z_{ref}, 99.9))$$

$$\hat{T} : FV \mapsto \hat{z} = \frac{FV - z_{min}}{z_{max} - z_{min}}$$

5. $\mathrm{BC}_{\mathrm{BETAVAE}}(o) = \hat{T} \circ FV(o)$

**Patch-BetaVAE**  Reinke et al. (2020) noticed that the $\beta$-VAE is not able to encode finer details and texture of patterns as the compression of the images to a 8-dimensional vector results in a general blurriness in the reconstructed patterns [62]. Therefore, we also implemented an additional variant denoted as PATCH-BETAVAE where the $\beta$-VAE is trained on "zoomed" $32 \times 32$ patches. A preprocessing step extracts the cropped patch around the image activation centroid $P : o \mapsto o[\bar{x}_m - 16 : \bar{x}_m + 16, \bar{y}_m - 16 : \bar{y}_m + 16]$. Then, the construction of $\mathrm{BC}_{\mathrm{PATCH\text{-}BETAVAE}}$ follows exactly the construction of $\mathrm{BC}_{\mathrm{BETAVAE}}$, except that the network architecture has only 3 convolutional layers instead of 6. Following the notations of the previous paragraph, $\mathrm{BC}_{\mathrm{PATCH\text{-}BETAVAE}}(o) = \hat{T} \circ FV \circ P(o)$ with $FV$ the pretrained model on image patches and $\hat{T}$ a normalizing function computed on $\mathcal{D}_{ref}$.

### B.1.2  Binning-based Diversity Metric

We follow existing approaches in the QD and IMGEP literature [55, 57] and use binning-based measure to quantify the diversity of a set of explored instances into a predefined BC space. The entire BC space is discretized into a collection of t bins $N_1, \ldots, N_t$ and the diversity is quantified as the number of bins filled over the course of exploration: $D_{|BC} = \sum_{i=1}^{t} \delta_i$ where $\delta_i = 1$ if the $N_i^{th}$ bin is filled, $\delta_i = 0$ otherwise.

We opt for a regular binning where each dimension of the BC space is discretized into equally sized bins. For all the results in the main paper, 20 bins per dimension are used for the discretization of the BC spaces. For recall, all the analytic BC spaces used in this paper are 8-dimensional and bounded in $[0, 1]^8$ (see previous section). This results in a total of $25.6 \times 10^9$ bins. Note however that for a given BC space, the maximum number of bins that can be filled by all possible Lenia patterns is unknown.

Because binning-based metrics directly depend on the choice of the bins discretization, we analyze in Figure 9 the impact of the choice of the number of bins on the final diversity measure. As we can see,

| Encoder | Decoder |
|---|---|
| Input pattern A: $256 \times 256 \times 1$ | Input latent vector z: $8 \times 1$ |
| Conv layer: 32 kernels $4 \times 4$, stride 2, 1-padding + ReLU | FC layers : 256 + ReLU, 256 + ReLU, $4 \times 4 \times 32$ + ReLU |
| Conv layer: 32 kernels $4 \times 4$, stride 2, 1-padding + ReLU | TransposeConv layer: 32 kernels $4 \times 4$, stride 2, 1-padding + ReLU |
| Conv layer: 32 kernels $4 \times 4$, stride 2, 1-padding + ReLU | TransposeConv layer: 32 kernels $4 \times 4$, stride 2, 1-padding + ReLU |
| Conv layer: 32 kernels $4 \times 4$, stride 2, 1-padding + ReLU | TransposeConv layer: 32 kernels $4 \times 4$, stride 2, 1-padding + ReLU |
| Conv layer: 32 kernels $4 \times 4$, stride 2, 1-padding + ReLU | TransposeConv layer: 32 kernels $4 \times 4$, stride 2, 1-padding + ReLU |
| Conv layer: 32 kernels $4 \times 4$, stride 2, 1-padding + ReLU | TransposeConv layer: 32 kernels $4 \times 4$, stride 2, 1-padding + ReLU |
| FC layers : 256 + ReLU, 256 + ReLU, FC: $2 \times 8$ | TransposeConv layer: 32 kernels $4 \times 4$, stride 2, 1-padding |

Table 1: $\beta$-VAE architecture used for $\mathrm{BC}_{\mathrm{BETAVAE}}$.

the ranking of the different IMGEP algorithms compared in Figure 5 of the main paper is invariant to this choice.

(a) Diversity of SLP in $BC_{\text{ELLIPTICAL-FOURIER}}$     (b) Diversity of TLP in $BC_{\text{LENIA-STATISTICS}}$

Figure 9: Influence of the choice of the number of bins on the diversity measure presented in Figure 5 of the main paper. The final diversity (number of occupied bins at the end of exploration, as shown in the y axis) is measured by varying the number of bins per dimension from 5 to 35. Results in the main paper use n=20 bins. Mean and std-deviation shaded area curves are depicted.

## B.2 Representational Similarity Analysis

We denote $\mathcal{D}_{ref}^{(small)}$ an external dataset of 3000 Lenia patterns (50% SLPs and 50% TLPs) which were collected with the same procedure than $\mathcal{D}_{ref}$.

Given two representations embedding networks $R_i$ and $R_j$ with 8-dimensional latent space, the RSA similarity index $RSA_{ij}$ is computed with the the linear Centered Kernel Alignment index (CKA) as proposed in [34]:

1. Compute the matrix of behavioral descriptors responses from each representation
   $Z_i = [R_i(o), o \in \mathcal{D}_{ref}^{(small)}] \in [0,1]^{3000 \times 8}$ and $Z_j = [R_j(o), o \in \mathcal{D}_{ref}^{(small)}] \in [0,1]^{3000 \times 8}$

2. Center the matrices responses:
   $Z_i \leftarrow Z_i - mean(Z_i, axis = 0)$ and $Z_j \leftarrow Z_j - mean(Z_j, axis = 0)$

3. $RSA_{ij} = CKA(Z_i Z_i^T, Z_j Z_j^T) = \frac{||Z_i \cdot Z_j^T||_F^2}{||Z_i \cdot Z_i^T||_F ||Z_j \cdot Z_j^T||_F}$
   where $||\cdot||_F$ represents the Frobenius norm

Representation Similarity Analysis (RSA) is used in Figure 3 of the main paper in two ways:

- To compare representations in *time*, i.e. where the the embedding networks $R_i$ and $R_j$ come from the same network but from different training stages

- To compare representations from different *modules* in HOLMES where the the embedding networks $R_i$ and $R_j$ are taken from the same time step (end of exploration) but from different networks.

## B.3 Human-Evaluator Selection of a *proxy*-BC for Evaluation of SLP and TLP Diversity

Relying on the external database $\mathcal{D}_{ref}$ of 15000 Lenia patterns (50% SLP - 50% TLP) and the SLP/TLP classifiers, we conducted an experiment with a human evaluator to select the analytic BC space that correlates the most with human judgement of what represents a *diversity of SLP* and a *diversity of TLP*.

The experiment consisted in repeatedly showing the human with two sets of patterns (as shown in Figure 10) and asking the human to click on the set that he considers is the more *diverse*, according to its intuitive notion of diversity. If the human cannot choose between the two sets, he can click on the "pass" button. In background, the procedure to generate the sets is the following:

1. Randomly select a (BC, category) pair, where BC ∈ {SPECTRUM-FOURIER, ELLIPTICAL-FOURIER, LENIA-STATISTICS, BETAVAE, PATCH-BETAVAE} and category ∈ {SLP,TLP}.

Figure 10: Interface used for selecting of the best proxy-BC analytic space that correlates with human judgement of what represents a *diversity of SLP* and a *diversity of TLP*.

2. Randomly draw 750 candidate sets of 6 images among the 7500 patterns of the current category.

3. Select the most *similar* set and the most *dissimilar* (i.e. diverse) set among those 750 sets. The (di-)ssimilarity of a set of 6 images is measured as a function of all the distances between each pair of images in the set, with distances being computed in the current BC space. This distance-based measure of diversity $D$, proposed in [65], measures the magnitude $M$ (dispersion) and variability $E$ (equability) of the set of S=6 points in the BC [65]:

$$M = \frac{S}{S-1} \sum_{i=1}^{S} \sum_{j=1}^{S} \frac{d_{ij}}{S^2}, \text{ where } d_{ij} \text{ is the pairwise euclidean distance}$$

$$E = \frac{1 + \sqrt{1 + 4H}}{2S}, \text{ where } H = \left[ \sum_{i=1}^{S} \sum_{j=1}^{S} \left( \frac{d_{ij}}{\sum_{i=1}^{S} \sum_{j=1}^{S} d_{ij}} \right)^2 \right]^{\frac{1}{1-2}}$$

$$D = 1 + (S-1) \times E \times M, M \in [0,1] and E \in [0,1]$$

This measure replaces the binning-based measure which can hardly be used here (as they are only 6 images most candidate sets are likely to fall in the same number of bins and be equally diverse).

4. The sets are displayed to the human in random presentation order.

This experiment was conducted with one human evaluator which performed a total of 500 clicks, i.e 50 times per (BC, category) pair. For each click per (BC,category) pair, the agreement score is 0 if the human selected the opposite set that the one considered as diverse by the BC, 0.5 is the human selected the "pass" button and 1 if the human selected the set considered as diverse by the BC. Table 2 reports the mean and standard deviation agreement scores of the human evaluator for each (BC, category) pair. The agreement score is significant at level $\alpha = 5\%$ if it is above $0.64 = 0.5 + 1.96 \times \sqrt{\frac{0.25}{50}}$.

Table 2: Human-evaluator agreement scores (mean $\pm$ std). Best scores are shown in bold.

|  | Spectrum-Fourier | Elliptical-Fourier | Lenia-Statistics | BetaVAE | Patch-BetaVAE |
|---|---|---|---|---|---|
| **SLP** | $0.5 \pm 0.18$ | $\mathbf{0.98 \pm 0.04}$ | $0.59 \pm 0.12$ | $0.1 \pm 0.06$ | $0.89 \pm 0.08$ |
| **TLP** | $0.2 \pm 0.13$ | $0.47 \pm 0.1$ | $\mathbf{0.92 \pm 0.07}$ | $0.75 \pm 0.08$ | $0.38 \pm 0.08$ |

As we can see, the human evaluator designated $BC_{\text{ELLIPTICAL-FOURIER}}$ as the best proxy space for evaluating the diversity of SLP (98% agreement score) and $BC_{\text{LENIA-STATISTICS}}$ as the best proxy space for evaluating the diversity TLP (92% agreement score). This is why those BCs are used in Figure 5 of the main paper.

## C  Experimental Settings

### C.1  Environment Settings

All experiments are done in the Lenia environment, as described in [6, 62]. As stated in the main paper, we use a $256 \times 256$ state size ($A \in \mathbb{R}^{256 \times 256}$) and a number of $T = 200$ steps for each run.

The $256 \times 256$ Lenia grid is a torus where the neighborhood is circular (i.e pixels on the top border are neighbors of the pixels on the bottom border and same between the left and right borders).

Lenia's update rule ($A^t \to A^{t+1}$) is defined as $A^{t+1} = A^t + \Delta_{\mathcal{T}} G(K * A^t)$, where:

- G defines a parametrized *growth mapping* function: exponential with $G(u; \mu, \sigma) = 2 \exp\left(-\frac{(u-\mu)^2}{2\sigma^2}\right) - 1$

- K defines a parametrized concentric-ring Kernel with:

$$K_C(r) = \exp\left(\alpha - \frac{\alpha}{4r(1-r)}\right), \text{ with } \alpha = 4 \quad \text{(Kernel core)}$$
$$K_S(r; \beta) = \beta_{\lfloor Br \rfloor} K_C(Br \bmod 1), \text{ with } \beta = (\beta_1, \beta_2, \beta_3) \quad \text{(Kernel shell)}$$
$$K = \frac{K_S}{|K_S|}$$

The update rule is therefore determined with 7 parameters:

- $R$: radius of the Kernel (i.e. radius of the local neighborhood that influences the evolution of each cell in the grid),
- $\mathcal{T} = \frac{1}{\Delta_{\mathcal{T}}}$: fraction of the growth update that is applied per time step,
- $\mu, \sigma$: growth center and growth width,
- $\beta_1, \beta_2, \beta_3$: concentring rings parameters that control the shape of the kernel.

See the project website `http://mayalenE.github.io/holmes/` for videos of the Lenia dynamics.

### C.2  Sampling of parameters $\theta$

The set of *controllable* parameters $\theta$ of the artificial agent include:

- The update rules parameters $[R, \mathcal{T}, \mu, \sigma, \beta_1, \beta_2, \beta_3]$
- CPPN-parameters that control the generation of the initial state $A^{t=1}$

For each exploration run, the IMGEP agent samples a set of parameters $\theta \in \Theta$ that generates a rollout $A^{t=1} \to \cdots \to A^{t=200}$.

Following the notations of Algorithm 1, there are two ways parameters $\theta \in \Theta$ are sampled:

1. During the $N_{init}$ initial runs, parameters are randomly sampled $\theta \sim \mathcal{U}(\Theta)$
2. During the goal-directed exploration runs, parameters are sampled from a policy $\theta \sim \Pi(\hat{BC}_i, \hat{g}, \mathcal{H})$. The $\Pi$ policy operates in two steps:
   (a) given a goal $g \in \hat{BC}_i$, select parameters $\hat{\theta} \in \mathcal{H}$ whose corresponding outcome is closest to $g$ in $\hat{BC}_i$
   (b) mutate the parameters by a random process $\theta = \text{MUTATION}(\hat{\theta})$

We therefore need to define 1) the random process $\mathcal{U}$ used to randomly initialize the parameters $\theta$ and 2) the random MUTATION process used to mutate an existing set of parameters $\hat{\theta}$.

Please note that for both we follow exactly the implementation proposed in Reinke et al. (2020) [62]. We therefore refer to section B.4 of their paper for a complete description of the implementation of the random initialization process and random mutation process. This includes the procedure used

for parameters that control the generation of the initial pattern $A^{t=1}$ and for parameters that control Lenia's update rule.

In this paper, we use the exact same hyperparameters as in [62] for initialization $\mathcal{U}$ and MUTATION of the CPPN-parameters that control the generation of the initial state $A^{t=1}$. We use slightly different hyper-parameters for the MUTATION of the parameters that control the generation of the update rule $[R, \mathcal{T}, \mu, \sigma, \beta_1, \beta_2, \beta_3]$, as detailed in table 3.

Table 3: Sampling of parameters for the update rule. The random initialization process $\mathcal{U}$ uses uniform sampling in an interval $[a, b]$. The random MUTATION is a Gaussian process $\theta = [\hat{\theta} + \mathcal{N}(\sigma_M)]_a^b$.

|  | $R$ | $\mathcal{T}$ | $\mu$ | $\sigma$ | $(\beta_1, \beta_2, \beta_3)$ |
|---|---|---|---|---|---|
| $[a, b]$ | $[2, 20]$ | $[1, 20]$ | $[0, 1]$ | $[0.001, 0.3]$ | $[0, 1]$ |
| $\sigma_M$ | 0.5 | 0.5 | 0.1 | 0.05 | 0.1 |

## C.3 Incremental Training of the BC Spaces

**Training Procedure** The networks are trained 100 epochs every 100 runs of exploration (resulting in 50 training stages and 5000 training epochs in total). The networks are initialized with *kaiming* uniform initialization. We used the Adam optimizer ($lr = 1\mathrm{e}{-3}$, $\beta_1 = 0.9$, $\beta_2 = 0.999$, $\epsilon = 1\mathrm{e}{-8}$, weight decay=$1\mathrm{e}{-5}$) with a batch size of 128.

**Training Dataset** The datasets are incrementally constructed during exploration by gathering the discovered patterns. One pattern every ten is added to the validation set (10%) and the rest is used in the training set (the validation dataset only serves for checking purposes and has no influence on the learned BC spaces). Importance sampling is used to give the newly-discovered patterns more weights. A weighted random sampler is used as follow: at each training stage t, there are $X$ patterns discovered so far among which $X_{new}$ have been discovered during the last 100 steps, we create a dataset $D_t$ of $X$ images that we construct by sampling 30% among the $X_{new}$ lastly discovered images and 70% among the $X - X_{new}$ old patterns. We also use data-augmentation, i.e at each training stage t, the images in $D_t$ are augmented online by random x and y translations (up to half the pattern size and with probability 0.6), rotation (up to 20 degrees and with probability 0.6), horizontal and vertical flipping (with probability 0.2), zooming (up to factor 3 with probability 0.6). The augmentations are preceded by spherical padding to preserve Lenia spherical continuity.

**IMGEP-VAE** The monolithic VAE architecture used in the IMGEP-VAE baseline is detailed in table 4. It has a total neural capacity of 2258657 parameters.

| **Encoder** | **Decoder** |
|---|---|
| Input pattern A: $256 \times 256 \times 1$ | Input latent vector z: $16 \times 1$ |
| Conv layer: 64 kernels $4 \times 4$, stride 2, 1-padding + ReLU | FC layers : 512+ ReLU, 512+ ReLU, $4 \times 4 \times 64$ + ReLU |
| Conv layer: 64 kernels $4 \times 4$, stride 2, 1-padding + ReLU | TransposeConv layer: 64 kernels $4 \times 4$, stride 2, 1-padding + ReLU |
| Conv layer: 64 kernels $4 \times 4$, stride 2, 1-padding + ReLU | TransposeConv layer: 64 kernels $4 \times 4$, stride 2, 1-padding + ReLU |
| Conv layer: 64 kernels $4 \times 4$, stride 2, 1-padding + ReLU | TransposeConv layer: 64 kernels $4 \times 4$, stride 2, 1-padding + ReLU |
| Conv layer: 64 kernels $4 \times 4$, stride 2, 1-padding + ReLU | TransposeConv layer: 64 kernels $4 \times 4$, stride 2, 1-padding + ReLU |
| Conv layer: 64 kernels $4 \times 4$, stride 2, 1-padding + ReLU | TransposeConv layer: 64 kernels $4 \times 4$, stride 2, 1-padding + ReLU |
| FC layers : 512+ ReLU, 512+ ReLU, FC: $2 \times 16$ | TransposeConv layer: 1 kernels $4 \times 4$, stride 2, 1-padding |

Table 4: VAE architecture used for IMGEP-VAE.

**IMGEP-HOLMES** For the IMGEP-HOLMES variant, the hierarchical representation starts with a single root module $R_0$ at the beginning of exploration. During each training stage, one node is *split* if it meets the following conditions:

- the reconstruction loss for that node reaches a plateau (running average over the last 50 training epochs is below $\epsilon = 20$)
- at least 500 patterns populate the node

- the node has not just been created (must has been trained for at least 200 epochs)
- it is not too early in the exploration loop (there must be at least 2000 patterns are explored)
- the total number of nodes in the hierarchy is below the maximum number allowed (we stop the expansion after 11 splits i.e. 23 modules)

Each time a split is triggered in a BC space node of the hierarchy $BC_i$, the boundary $\mathcal{B}_i$ is fitted in the latent space as follows: K-Means algorithm with 2 clusters is ran on the patterns that currently populate the node. The resulting clusters are kept fixed for the rest of the exploration, therefore when a pattern is projected in the split node, it is sent to the left children if it belongs to the first cluster on the latent space and to the right children otherwise.

For the IMGEP-HOLMES variant, the final hierarchy has a total of 23 VAE modules. The architecture is identical for each module and is detailed in table 5. At the end of exploration, HOLMES has a total neural capacity of 2085981 parameters. Each base module VAE has a capacity of 86225 parameters and connections of 4673 parameters ($2085981 = 23 \times 86225 + 22 \times 4673$).

**Encoder**

| |
|---|
| Input pattern A: $256 \times 256 \times 1$ |
| Conv layer: 16 kernels $4 \times 4$, stride 2, 1-padding + ReLU |
| Conv layer: 16 kernels $4 \times 4$, stride 2, 1-padding + ReLU |
| Conv layer: 16 kernels $4 \times 4$, stride 2, 1-padding + ReLU                 **lf_c:** 16 kernels $1 \times 1$, stride 1, 1-padding |
| Conv layer: 16 kernels $4 \times 4$, stride 2, 1-padding + ReLU |
| Conv layer: 16 kernels $4 \times 4$, stride 2, 1-padding + ReLU |
| Conv layer: 16 kernels $4 \times 4$, stride 2, 1-padding + ReLU |
| FC layers : 64+ ReLU, 64+ ReLU, FC: $2 \times 16$ |

**Decoder**

| |
|---|
| Input latent vector z: $16 \times 1$ |
| FC layers : 64+ ReLU,                               **gfi_c:** 64+ReLU |
| FC layers: 64+ ReLU, $4 \times 4 \times 16$ + ReLU |
| TransposeConv layer: 16 kernels $4 \times 4$, stride 2, 1-padding + ReLU |
| TransposeConv layer: 16 kernels $4 \times 4$, stride 2, 1-padding + ReLU |
| TransposeConv layer: 16 kernels $4 \times 4$, stride 2, 1-padding + ReLU  **lfi_c:** 16 kernels $1 \times 1$, stride 1, 1-padding |
| TransposeConv layer: 16 kernels $4 \times 4$, stride 2, 1-padding + ReLU |
| TransposeConv layer: 16 kernels $4 \times 4$, stride 2, 1-padding + ReLU |
| TransposeConv layer: 1 kernel $4 \times 4$, stride 2, 1-padding       **recon_c:** 1 kernel $1 \times 1$, stride 1, 1-padding |

Table 5: Module architecture used for IMGEP-HOLMES. All the modules $R_i$ have this architecture for the base VAE network as well as the connections (except $R_0$ which does not have the connections).

# D  Additional Results

## D.1  RSA complete temporal analysis and statistics

Figure 11 shows how HOLMES is able to progressively builds a hierarchy of behavioral characterization spaces from the incoming data. The data is here collected by the IMGEP-HOLMES algorithm, with 50 training stages occurring each 100 steps. At start (stage 0), the hierarchy contains only the root node at the top of the figure (BC 0). Node saturation occurs at training stages where the RSA similarity index between the representation at that stage and the representations at all subsequent stages is high (yellow). For example, we see on the figure that the root node saturates after approximately 15-20 training stages. When a node saturates, HOLMES splits it in two child nodes (see section A.1 for details on the splitting procedure). For example, the root node BC 0 is split into the child nodes BC 00 and BC 01 at training stage 21, as indicated by the fact that the RSA plots of

Figure 11: Example of a hierarchy of behavioral characterization spaces learned by HOLMES. It starts with a root node (BC 0, top) and iteratively splits the learned latent spaces, resulting in a tree structure (with leaf nodes at the bottom). In each node, we display the RSA similarity index between 0 (dark blue, not similar at all) and 1 (yellow, identical), where representations are compared in time between the different training stages.

these child nodes start at that stage. When a node is split, the parent node is frozen and learning only continues in leaf nodes (as indicated by the RSA indexes of a parent node being all at 1 after a split). We observe that some child nodes saturate much more quickly than others. For example, node BC 000 saturates only a few training stages after its split from BC 00, while its sibling BC 001 never saturates until the end of the training at stage 50. The RSA analysis of node BC 001 indeed shows that the learned representation continues to evolve as training occurs. This means that this node corresponds to a part of the BC space constituting a rich progress niche for the base module VAE associated with that node. In contrast, node BC 000 will require further splitting to discover such progress niches in its child nodes. The reader can refer to Figure 15 in section F for visualizing discovered patterns in each nodes of the hierarchy.

Figure 12: RSA similarity index between 0 (dark blue, not similar at all) and 1 (yellow, identical). (Top) RSA matrix for one experiment as shown in Figure 3 of main paper. (Bottom) statistics over the 10 repetitions: (bottom-VAE) RSA index similarity between representations coming from two consecutive training stages (mean and std); (bottom-HOLMES): histogram of RSA index similarity between all pairs of modules in HOLMES (aggregated over the 10 repetitions).

Figure 12 complements Figure 3 of the main paper with statistical results over 10 repetitions. The statistical results (bottom row) confirm our analysis: the VAE representation saturates quite early in the exploration loop and the representations learned by HOLMES modules are dissimilar from one module to another. Indeed we can see that the VAE representations of all experiments saturate after 15 training stages (high RSA $\approx 1$ between remaining consecutive training stages). The histogram of similarity index between all pairs of modules in HOLMES (for all experiments) show a concentration between [0,0.3] (i.e. very low similarity).

## D.2 Additional IMGEP baselines with a monolithic BC space

In section 4.2 of the main paper, we compared the incremental training of behavioral training between an IMGEP equipped with a monolithic VAE (IMGEP-VAE) and an IMGEP equipped with the hierarchy of VAEs (IMGEP-HOLMES).

**Baselines** In this section, we consider different baselines for the training strategy of the monolithic architecture: **BetaVAE** [5], **BetaTCVAE** [9], **TripletCLR** [8, 67], **SimCLR** [11] and **BigVAE**. All the baselines have the same encoder architecture and training procedure than the main baseline IMGEP-VAE (as detailed in section C.3). The baselines differ in their approach to train the encoder network, including several variants of variational-autoencoders and contrastive approaches.

The first two variants BetaVAE [5] and BetaTCVAE [9] build on the VAE framework and augment the VAE objective with the aim to enhance interpretability and disentanglement of the latent variables. Therefore only the training loss of the VAE (see section A.1) differs:

- The BetaVAE objective re-weights the $b$ term by a factor $\beta > 1$:
  $$\mathcal{L}_{\text{BETAVAE}}(\theta, \phi; \mathbf{x}, \mathbf{r}) = \underbrace{\mathbb{E}_{\hat{p}(\mathbf{x})}\left(\mathbb{E}_{q_\phi(\mathbf{r}|\mathbf{x})}\left(-\log p_\theta(\mathbf{x}|\mathbf{r})\right)\right)}_{a} + \beta \times \underbrace{\mathbb{E}_{\hat{p}(\mathbf{x})}\left(D_{\text{KL}}\left(q_\phi(\mathbf{r}|\mathbf{x})||p(\mathbf{r})\right)\right)}_{b}$$
  Our baseline uses $\beta = 10$.

- The BetaTCVAE objective augments the VAE objective with an additional regularizer that penalizes the *total correlation* (dependencies between the dimensions of the representation):

$$\mathcal{L}_{\text{BETATCVAE}}(\theta, \phi; \mathbf{x}, \mathbf{r}) = \mathbb{E}_{\hat{p}(\mathbf{x})} \left( \mathbb{E}_{q_\phi(\mathbf{r}|\mathbf{x})} \left( -\log p_\theta(\mathbf{x}|\mathbf{r}) \right) \right) +$$

$$\alpha \times \underbrace{I_{q_\phi}(\mathbf{x}|\mathbf{r})}_{mutual\ information} + \beta \times \underbrace{TC(q_\phi(\mathbf{r}))}_{total\ correlation} + \gamma \times \underbrace{\sum_j D_{KL}(q_\phi(z_j)||p(z_j))}_{elementwise\ KL}$$

  Because TC is not tractable, they [9] propose two methods based on importance sampling to estimate it: *minibatch weighted sampling* (mws) and *minibatch stratified sampling (mss)*. Our baselines uses $\alpha = 1$, $\beta = 10$, $\gamma = 1$ and $mss$ importance sampling.

The second two variants TripletCLR [8, 67] and SimCLR [11] use contrastive approaches as training strategy for the encoder. Contrary to the VAE variants, these approaches drop the decoder networks and pixel-wise reconstruction as their training objective operates directly in the latent space. The encoders are trained to maximize agreement between differently augmented versions of the same observation $o$. We used to 2 variants for the contrastive loss:

- Triplet Loss: $\mathcal{L}_{\text{TRIPLETCLR}}(A, P, N) = \max\left(d(R(A), R(P)) - d(R(A), R(N)) + \alpha, 0\right)$ where $R$ is the embedding network, $A$ is an anchor input (pattern $o$ in the training dataset), $P$ is the positive input (augmented version of $o$), $N$ is the negative input (other pattern $o'$ randomly sampled in the training dataset), $d(\cdot, \cdot)$ is the distance in the latent space (we use cosine similarity) and $\alpha$ is a margin between positive and negative pairs (we use $\alpha = 1$)

- SimCLR Loss: $\mathcal{L}_{\text{SIMCLR}}(A, P) = -\log \frac{\exp sim(z_A, z_P)/\tau}{\sum_N \mathbb{1}_{[N \neq A]} \exp sim(z_A, z_N)/\tau}$ where $\tau$ denotes the temperature parameter (we use $\tau = 0.1$); $sim$ the similarity distance (we use cosine similarity); and $z$ represent the latent features onto which operates the contrastive loss. Please note that for this variant the encoder is coupled to a *projection head* network $g(\cdot)$ such that $o \xrightarrow{R} r \xrightarrow{g} z$ (We use g: FC 16→16 + RelU, FC 16→16). Here the positive pair $(A, P)$ is contrasted with all negative pairs $(A, N)$ in the current batch. The final loss is computed across all positive pairs in a mini-batch.

Finally the BigVAE baseline uses the same architecture and training strategy than the main VAE baseline, but with a much larger embedding capacity (368 dims instead of 16 dims) corresponding to the total embedding capacity of HOLMES if we would concatenate its 23 BC latent spaces.

**Results**   The results are summarized in Figure 13 and corroborate with the insights in the main paper.

- Lack of *plasticity* for the all VAE variants, i.e. inability to adapt the learned features to novel niches of patterns. The bias that we observed in the main paper is confirmed in the RSA analysis, even when changing the training objective or the encoding capacity. Interestingly, IMGEP-BetaVAE and IMGEP-BetaTCVAE show the same profile of discovered diversities than VAE (good at finding a diversity of SLPs but bad for TLPs) whereas the IMGEP-BigVAE seems to have a reversed bias (good at finding a diversity of TLPs but bad for SLPs). We attribute this effect to the difficulty of VAEs with low embedding capacity to capture textures with fine-grained structures (i.e. TLPs) whereas when given a higher encoding-capacity they can more accurately represent TLPs. Therefore the variants with small capacity representations seem better suited for exploring diverse SLPs (to the detriment of TLPs) whereas BigVAE seem better suited for exploring diverse TLPs (to the detriment of SLPs).

- Lack of *stability* for all the contrastive variants, where features are drastically different from one training stage to the other. Contrary to the VAE variants, those approaches do not exhibit a strong *bias* in their BC and therefore do not seem to differ much from the *default* diversity found in Lenia (represented with the black curve), at least for the two *types* of diversity measured in Figure 13.

Figure 13: This figure complements the results presented in the main paper, where we replace the baseline with the monolithic BC space (IMGEP-VAE) with different architectures and training strategies. Each row is a baseline denoted as IMGEP-X (where X represents the training strategy used for training the monolithic representation). For each row, we display: (left) RSA matrix where representations are compared in time between the different training stages, as shown in Figure 3 of the main paper; (middle-right) exact same plots than Figure 5 of the main paper where we replace the monolithic IMGEP-VAE baseline (pink curve) by the other baseline (of the current row). Therefore only the pink curve vary between the different graphs of one column.

## D.3  Ablation Study: Impact of the Lateral Connections

Figure 14: RSA Analysis of the effect of the *lateral connections* on the ability for HOLMES to learn *diverse* module BCs. Each row is an ablation experiment with the corresponding connection scheme. (Left) RSA matrix for one experiment repetition (seed 0), with similarity index between 0 (dark blue, not similar at all) and 1 (yellow, identical). Representations are compared at the end of exploration between the different modules (ordered by their creation time on the left-to-right x-axis). (Right) Histogram of RSA index similarity between all pair of modules (aggregated over the 10 repetitions).

We conducted 5 ablation experiments of the IMGEP-HOLMES variant presented in the main paper, each has 3 repetitions with different seeds. Each ablation experiment considered a different connection scheme with either zero or only one connection among **lf_c**, **gfi_c**, **lfi_c** and **recon_c** (proposed connections in HOLMES, see Figure 6 and Table 5). As shown in Figure 14, the lateral connections are *essential* to learn *diverse* behavioral characterizations among the different modules of the hierarchy. Indeed we can see that IMGEP-HOLMES without any connection (first row in the figure) learns BCs that are highly similar from one module to another (histogram concentrated around [0.8, 1] RSA indexes, i.e. very similar). We can also see that connections toward the last layers of the decoder are seemingly the more important (lfi_c and recon_c) as IMGEP-HOLMES with only one of such connection succeeds to learn dissimilar BCs per module (the histogram is shifted toward lower RSA indexes). However, the connection at the encoder level (lf_c) and close to the embedding level (gfi_c) seem less necessary, or at least alone are not sufficient to allow HOLMES modules escaping the bias inherent to the VAE learning (show a similar histogram of RSA indexes than the no-connection variant). The connection scheme used in the main paper (last row in the figure) seems to be the best suited to learn diverse BCs (histogram concentrated around [0.8, 1] RSA indexes, i.e. very dissimilar).

# E    Comparison of HOLMES with Related Methods

In this section we provide a comparison of HOLMES with recent work in the literature: CURL [61], CN-FPM [40] and pro-VLAE [45]. Those approaches also propose to dynamically expand the network capacity of a VAE in the context of *continual* representation learning, and therefore share similarities with HOLMES. In table 6, we provide a high-level comparison of the proposed approaches which compare the different architectures according to their *structural bias*, handling of *catastrophic forgetting*, architecture for *dynamic expansion*, handling of *transfer* between the different group of features, criteria for the *expansion trigger*, and if they are performing *data partitioning* (i.e. learn different set of features for different niches of observations).

Table 6: High-level comparison of the general choices of HOLMES with those of previous methods: CURL [61], CN-FPM [40] and pro-VLAE [45]. Please refer to the original papers for more details.

|  | CURL | CN-DPM | pro-VLAE | HOLMES |
|---|---|---|---|---|
| **Structural Bias** | Mixture of Gaussians (in a single VAE latent space) | Dirichlet Process Mixture (flat set of VAE modules) | Hierarchical Levels (in a single VAE) | Hierarchical Mixture (binary tree of VAE modules) |
| **Catastrophic Forgetting** | Generative Replay | Freeze | "fade-in" coefficient (same network) | Freeze |
| **Dynamic Expansion** | New component in the MoG | New module VAE | New feature layer in the VAE | New module VAE |
| **Transfer** | Single Shared Network with several "heads" | Lateral connections (exhaustive as in PNN [64])) | Single Shared Network with several "levels" | Lateral connections (parent-to-children only) |
| **Expansion Trigger** | Short-Term Memory Size | Short-Term Memory Size | Predetermined | Node Saturation |
| **Data Partitioning** | Soft partitioning | Soft partitioning (coupling of each VAE with discriminator) | None (same network) | Hard Partitionning (boundary in $BC_i$) |

As we can see, while HOLMES shares *conceptual* ideas with those approaches, our approach has key differences:

1. It uses a hierarchy of different latent spaces whereas CURL uses a single latent space, CN-DPM uses a flat set of different latent spaces and pro-VLAE uses a fixed-set of latent spaces (different levels in one network)

2. CURL and CN-DPM show results in the context of continual multi-task classification and demonstrate that their modular architecture can separate well the latents allowing to unsupervisedly discriminate between the different input observations / tasks (eg: discriminate digits in MNIST at test time when they have been sequentially observed at train time). However, CURL does not use different features for the different niches of observations and it is not clear if the flat approach of CN-DPM does learn different features between the different modules. However HOLMES targets to learn dissimilar set of features per BC in order to achieve *meta-diversity*.

3. Pro-VLAE is not applied in the context of continual learning but rather proposes to progressively learn features at different levels in the VAE layers, showing that it can successfully disentangle the features. Even though disentanglement is a key property to avoid redundant features, we believe that it is also key to have diverse set of features for the different niches of observed instances.

# F    Additional Visualisations

Figure 15 shows examples of patterns discovered by IMGEP-HOLMES (non-guided) within the learned tree hierarchy. The patterns shown in the root node are representative of the diversity of all the discovered patterns in that particular run (100% of the patterns). The boundaries fitted when splitting each non-leaf node (see procedure in section A.1) makes each pattern follow a particular path in the hierarchy, from the root node to a leaf node. Goals are sampled by the IMGEP by first sampling a leaf uniformly among all existing leafs, then sampling uniformly in the hypercube fitted around currently reached goals within that leaf (see section 3.2 of the main paper). However, we observe that the percentage indicated in each node does not reflect this uniformity (for example, only 5.7% of the patterns fall in the leaf BC 001). The interpretation is that leafs with low percentages correspond to unstable niches: when a goal is sampled in such a leaf, the small mutation applied in the parameter-sampling policy is sufficient to produce a pattern which is different enough to fall in another leaf.

We qualitatively observe in Figure 15 that the boundaries fitted during the splitting procedure tend to separate the patterns into visually distinct categories. For example, the proportion of TLPs is much higher in BC 01 compared to BC 00 ; the leaf BC 00000 contains only blank patterns while its sibling BC 00001 contains only SLPs ; the nodes below BC 01111 (bottom-right of the tree) contains only TLPs.

Figures 16 and 17 show discovered patterns when IMGEP-HOLMES is guided towards SLPs or TLPs, respectively, through simulated user feedback as described in section 4.3 of the main paper. We observe that the user guidance is able to dramatically affect both the diversity of the discovered patterns and the structure of the hierarchy . When guided towards SLPs, most of the discovered patterns are SLPs (most TLPs in Figure 16 are concentrated in the leafs BC 01110 and BC 01111 which represent approximately 15% of the discovered patterns). On the contrary, when guided towards TLPs, most of the discovered patterns are TLPs (most SLPs in Figure 17 are concentrated in the leafs BC 000 and BC 001 which represent approximately 34% of the discovered patterns). As a consequence of this bias toward either SLPs or TLPs, we observe that HOLMES has created more branches in the direction of the desired patterns (either SLPs or TLPs) in order to enrich their corresponding representations.

Additional visualisations can be found on the project website `http://mayalenE.github.io/holmes/`.

Figure 15: Examples of patterns discovered by IMGEP-HOLMES (non-guided) within the learned tree hierarchy. The hierarchy is the same as in Figure 11. In each node is displayed the percentage of discovered patterns directed through that node, as well as a set of pattern images representative of the diversity within that node (the set is built with the procedure described in section B.3). The number of patterns per node reflects the indicated percentage.

Figure 16: Examples of patterns discovered by IMGEP-HOLMES within the learned tree hierarchy, when guided towards SLPs through simulated user feedback as described in section 4.3 of the main paper. Same convention as in Figure 15.

Figure 17: Examples of patterns discovered by IMGEP-HOLMES within the learned tree hierarchy, when guided towards TLPs through simulated user feedback as described in section 4.3 of the main paper. Same convention as in Figure 15.