[Reviews · NeurIPS 2020]

Review 1

Summary and Contributions: This paper is concerned with the automated discovery of interesting patterns in morphogenetic systems. It introduces the idea of meta-diversity search and shows that a dynamic and modular neural architecture allows for a more controllable search for diversity. =============== After rebuttal ================== The rebuttal addresses part of my issues and the new overview figure should make it easier for the reader to follow. Therefore I'm changing my score from a 7 to an 8. However, for a strong accept, I think the authors would need to apply their approach to at least one other (ideally functional) domain. I'm very much looking forward to the next steps in this research direction.

Strengths: The work introduces important new methods to controlling self-organizing morphogenetic systems. It combines a variety of different techniques such as exploratory search methods, the automatic and incremental discovery of different behavioral characterisation spaces, and model architecture. The results show that the system allows to quickly adapt the type of created diversity to a simulated end-user.

Weaknesses: Because of the diversity of different techniques, it was initially difficult to understand the paper's contributions and how everything fits together. The abstract and introduction could be updated to further help the reader understanding the paper's most important contributions. Additionally, a more high-level overview figure than the one presented in Fig. 1 could help to convey how all the different parts fit together. The second weakness is that the system is currently only applied to one particular domain. It would be very interesting to see how the approach would scale to a domain that requires functional content, such as the self-organization of robot morphologies.

Correctness: The claims made in the paper appear correct.

Clarity: The introduction could make the important contributions of the paper and how the different parts fit together more clear.

Relation to Prior Work: Prior work is described in succinct but in sufficient detail. Work on involving user-feedback, such as interactive evolution, could be added as well, if space permits.

Reproducibility: Yes

Additional Feedback:


Review 2

Summary and Contributions: This paper gives interesting evidence for the need of modularity for better exploration and diversity in goal-driven systems. In particular, the papers shows that a single neural network is worse than a dynamic+modular architecture of sub-networks on diversity metrics. The paper formulates the notion of meta diversity, a search algorithm to hierarchically (tree structured) build modules that route decisions to either create new networks to handle incoming observations or decode it outcomes. A continuous version of the game-of-life system is used (Lenia) as the experimental platform. In this system, different starting states, rules and interventions lead to vastly different outcomes. The diversity metrics are measured on such outcomes. Additionally a human user can guide the system's diversity in particular directions by intervening on the decision boundary between different sub modules. This enables a subjective notion to the exploration process. ---------- After rebuttal ------------ The rebuttal clarifies most of the concerns raised during the review process. For this reason I am increasing my score for this paper.

Strengths: This paper studies a very interesting, complex and dynamical test system with a large and intuitive space for exploration. The outcomes from such a dynamical system is combinatorial but has an ecological and intuitive basis -- making it an interesting test bed to study the influence of subjective interventions on exploration. This paper has also interesting implications for continual learning of neural networks in the reinforcement learning context. This paper is related to the progressive neural networks paper but goes beyond it by learning a tree-structured hierarchy of modules. It would have been interesting to see how the structure between modules affects exploration outcomes. How would the original progressive neural networks like topology for the modules work?

Weaknesses: "Because β-VAE can poorly represent high-frequency patterns, another variant trained on cropped patches is proposed" - patch beta VAE is introduced without any explanation or motivation in the main text How does this method compare to all other species that have been discovered in Lenia by others methods or manual interventions? Is there a metric to evaluate novelty? Why is a monolithic VAE better for diversity in some cases but for others? "when their existence remained an open question raised in the original Lenia paper" - is there a quantitative novelty metric to validate this claim?

Correctness: The claims and methods look accurate and reasonable.

Clarity: - The text to explain how the goal space interacts with Sec 3.1 can be more simply explained. It would be very helpful to pictorially show the exploration process to get better/faster intuition - Section 3.2 is dense and having simple intuitive explanations for each step would go a long way. For instance, 1) how is the parameter mutation process done, 2) why are particular goal sampling strategies chosen over others and what is the general space here? - The paper is currently dense and hard to follow. The underlying proposal and experiments are very interesting but it is difficult to understand due to a large number of unexplained choices and moving parts.

Relation to Prior Work: I would like to better understand what kind of diversity other approaches have produced in the Lenia system.

Reproducibility: Yes

Additional Feedback:


Review 3

Summary and Contributions: The paper proposes a novel way of exploratory search that combines hierarchical clustering with goal-based intrinsic motivation techniques in a learned latent space for finding a diverse set of morphogenetic systems. The automation of diversity-driven discovery is an important area of research as human-based manual tuning is too expensive and slow. The novel systems [e.g. 50] tries to mitigate it by introducing human-made representation or learned representation (e.g. with VAE) and automate diversity-driven discovery in this space. The authors proposed to make one more step of automation and learn representations mapping that are diverse calling this meta-diversity search and showed that this search leads to diverse clusters of patterns. Main contributions: - The novel objective of meta-diversity search and architecture suitable for learning organized (possibly guided by human preference) hierarchical representations. - Empirical evaluation of diversity of the discovered patterns and possible guidance by a human. === after rebutal === Thank you for the informative response. No major concerns remain from my side.

Strengths: - Novel formulation of the diversity that aims to discover diverse clusters of patters. - Empirical evaluation of the previously proposed approaches to show that they discover patterns that are diverse only in one behavioral characterization (BC). - As the task is not fully specified without human preferences, the authors proposed to use human guidance to search patterns that are more interesting for people.

Weaknesses: - As the main novelty lies in the clustering of different patterns, the usage of K-means in embedding space G could be explained better. Maybe also a discussion of other possible choices and their comparison would be helpful. - As several previous approaches were also restricted only to Continuous Game of Life, another application could make the main topic of diversity search (and meta-diversity search) more task-independent. For example, it is not clear how useful the same clustering approach would help in different complex dynamical systems.

Correctness: yes

Clarity: The paper is well-written, however, the main novel clustering part (3.1) of the paper should be explained and described better.

Relation to Prior Work: yes

Reproducibility: Yes

Additional Feedback: The webpage with the additional images and explorer for all the generated patterns is really nice.

[Author Response · NeurIPS 2020]

**Paper ID 9804 - Hierarchically-Organized Latent Modules for Exploratory Search in Morphogenetic Systems**

We thank reviewers (named R2-R3-R4) for their valuable feedback. We are glad they shared a great interest in the research problem and in the experimental framework, highlighting the relevance of the targeted complex dynamical systems (R3) as well as the importance for novel methods for automating discovery in those (R2,R3,R4). We are pleased that our evaluation and interactive visualisations successfully conveyed the limitations of existing approaches and the relevance of the proposed meta-diversity objective (R4). We are also encouraged that our modular approach for incremental learning of diverse organized representations was considered novel, more efficient than monolithic architectures and more suitable for integrating human-guided interventions (R2,R3,R4) which was recognized as necessary to fully specify the exploratory task (R4), and with broader interesting implications for continual learning scenarios (R3). We answer reviewers comments below and will incorporate all feedback in the the final paper version.

**Paper organization & Clarity** One primary concern of reviewers was the difficulty of grasping how the different components of our method fit together. Especially, it was suggested to improve section 3 (R3,R4) and to redo the main figure with a more high-level overview (R2,R3). We agree and propose to update the paper as follows:
1) Replace Figure 1 (see on the right) to integrate both the representation learning part (hierarchical clustering, section 3.1) and the exploration part (goal-based intrinsic motivation, section 3.2). Please zoom in for full-size view.

Figure 1: IMGEP-HOLMES overview

2) Simplify section 3.2 with intuitive explanations of the different steps (now-illustrated from 1 to 5 in the figure), as suggested by R3. Moreover, shortening 3.2 will free up some space for the following proposed updates.
3) Emphasize the clustering of the different patterns in section 3.1, which we agree is central (R4). There are two main choices: the training strategy that determines the latent distribution of patterns (we use VAEs) and the clustering algorithm itself (K-means). We recall that *"the genericity of HOLMES architecture [...] allows many other design choices to be considered in future work"* (last sentence of 3.1) and that our choices are discussed in appendix A.1.
4) Add a paragraph in section 5 with prior work on interactive exploration of patterns (R2) such as:
Langdon (2005) "Pfeiffer – A distributed open-ended evolutionary system", AISB ;
Secretan et al. (2011) "Picbreeder: A case study in collaborative evolutionary exploration of design space", Evolutionary Computation.

**Comparison of the different methods in Lenia** R3 expressed his interest to *"better understand what kind of diversity other approaches have produced in the Lenia system"*. First, evaluating the *kind* of diversity that is produced in a complex system like Lenia raises important questions, and we hope to have contributed in 3 ways with (i) several quantitative diversity metrics (relevant from the point of view of their novelty criteria); (ii) empirical evaluations with several visualisations and interactive web-interfaces and (iii) hybrid evaluations that rely on a human evaluator to select meaningful quantitative metrics (as proposed in section 4.3 and detailed in appendix B.3). Secondly, as we are ultimately evaluating diversity, our baselines are diversity-driven algorithms (population-based IMGEPs as in [50] that are similar to Novelty Search) that mainly differ in the BC space definition and for which we compared several hand-defined and unsupervisedly learned variants in section 4.1 and appendix D.2. In addition we also compare with a random exploration approach, and while we can hardly compare quantitatively with manual approaches from [6,7] that only keeps few "interesting" discoveries, our project website provides several videos and links to visualise those manually-identified discoveries - together with the full database of the automatically-identified discoveries of all the considered baselines. Additional clarifications asked by R3 are hopefully addressed below:
- Our claims on the novelty of the identified *pattern-emitting* behaviors are based on [7] which states the *"open questions raised in the original Lenia paper (Chan, 2019): Do self-replicating and pattern-emitting lifeforms exist in Lenia?"*
- The chosen analytic BC spaces (including PatchBetaVAE) are fully motivated and described in appendix B.1.1 but not focused in the main paper as many others could be envisaged. Similarly, our intuitions on why monolithic VAEs are better suited for exploring one type of diversity but not the other can be found in appendix D.2.
- The original flat-topology PNN architecture assumes a predefined sequence of tasks and cannot, as-it-is, be applied to our problem: it will need to autonomously handle the clustering of the incoming data distribution and dynamically expand accordingly over its lifetime. However, we refer to section E in Appendix for a comparison with other recently-proposed architectures that also share conceptual similarities with the original PNN architecture.

**Generalisation to other domains** As stated by R3, Lenia can produce combinatorial outcomes with appearances and dynamics comparable to real-world biology making it a very interesting and intuitive test-bed. Yet, we agree with R2 and R4 and are currently working on applying the proposed meta-diversity search framework to explore other complex systems: (i) the morphological affordances of bio-inspired robot designs and (ii) the behavior-space of real "wet" systems for which bio-chemists still lack of an intuitive understanding, such as the oil-droplet system studied in [19].

[Meta-Review · NeurIPS 2020]

This paper proposes a hierarchical method for representation learning and goal-directed search in morphogenetic systems, and is evaluated on a particular type of cellular automata (Lenia). This method allows for identifying diverse and “interesting” regions of space in the dynamical system, also supporting small amounts of human feedback to identifying preferred regions of space. R1 and R4 praised the novelty of this approach, with R3 also finding it interesting and highlighting its implications for other areas of research. The reviews initially had some concerns with clarity, but these were satisfactorily addressed by the rebuttal. Another issue, highlighted by R1 and R4, was that the system has only been evaluated on a single dynamical system, and so its applicability to other domains is unclear. I found the paper quite interesting and unique, and believe it will be very thought-provoking and interesting to the NeurIPS community. While I agree that it would be good to see a demonstration of the system in other domains, the paper as it stands is an important contribution in and of itself, with many new ideas. I therefore recommend acceptance.